# K-Anonymity Privacy Protection Algorithm for Multi-Dimensional Data against Skewness and Similarity Attacks

**DOI:** 10.3390/s23031554

**Published:** 2023-01-31

**Authors:** Bing Su, Jiaxuan Huang, Kelei Miao, Zhangquan Wang, Xudong Zhang, Yourong Chen

**Affiliations:** 1School of Computer and Artificial Intelligence, Changzhou University, Changzhou 213164, China; 2College of Information Science and Technology, Zhejiang Shuren University, Hangzhou 310015, China

**Keywords:** K-anonymity, multi-dimensional data, skewness attack, similarity attack, privacy protection

## Abstract

Currently, a significant focus has been established on the privacy protection of multi-dimensional data publishing in various application scenarios, such as scientific research and policy-making. The K-anonymity mechanism based on clustering is the main method of shared-data desensitization, but it will cause problems of inconsistent clustering results and low clustering accuracy. It also cannot defend against several common attacks, such as skewness and similarity attacks at the same time. To defend against these attacks, we propose a K-anonymity privacy protection algorithm for multi-dimensional data against skewness and similarity attacks (KAPP) combined with *t*-closeness. Firstly, we propose a multi-dimensional sensitive data clustering algorithm based on improved African vultures optimization. More specifically, we improve the initialization, fitness calculation, and solution update strategy of the clustering center. The improved African vultures optimization can provide the optimal solution with various dimensions and achieve highly accurate clustering of the multi-dimensional dataset based on multiple sensitive attributes. It ensures that multi-dimensional data of different clusters are different in sensitive data. After the dataset anonymization, similar sensitive data of the same equivalence class will become less, and it eventually does not satisfy the premise of being theft by skewness and similarity attacks. We also propose an equivalence class partition method based on the sensitive data distribution difference value measurement and *t*-closeness. Namely, we calculate the sensitive data distribution’s difference value of each equivalence class and then combine the equivalence classes with larger difference values. Each equivalence class satisfies *t*-closeness. This method can ensure that multi-dimensional data of the same equivalence class are different in multiple sensitive attributes, and thus can effectively defend against skewness and similarity attacks. Moreover, we generalize sensitive attributes with significant weight and all quasi-identifier attributes to achieve anonymous protection of the dataset. The experimental results show that KAPP improves clustering accuracy, diversity, and anonymity compared to other similar methods under skewness and similarity attacks.

## 1. Introduction

With the arrival of the big data era, the amount of digital information continues to surge. The analysis, mining, and application of massive data have attracted great attention to governments, industries, and research departments, etc [1]. It is not rare that health sectors and hospitals may share patient details with organizations such as research institutions for further analysis [2,3]. Although data sharing has given us convenience, it may also bring challenges in privacy and ethics. The published datasets typically contain large amounts of multi-dimensional sensitive data. That is to say, each individual has multiple sensitive information (e.g., shopping habits, medical history, and driving records) [4]. Attackers can use public data to analyze sensitive personal information, which may result in privacy disclosure. It is necessary for both data owners and data publishers to protect privacy in data publication and data use [5,6,7]. Thus, a publicly published privacy protection method for multi-dimensional data is needed to prevent the privacy disclosure of multiple sensitive attributes.

The privacy protection methods for multi-dimensional data, including encryption, difference privacy, K-anonymity, and so on [8]. The encryption methods have high computational complexity and are unsuitable for the encryption of extensive multi-dimensional data. Moreover, they cannot guarantee the privacy security of multi-dimensional data-sharing receivers. The difference privacy methods need to add a large amount of randomization into the structure, and it will cause a decline in the availability of multi-dimensional data. The K-anonymity methods use some methods (e.g., generalization, clustering, decomposition, and replacement) to anonymize multi-dimensional data before data sharing and have attracted much attention in recent years. In K-anonymity algorithms, a certain amount (≥*k*) of shared data is required to make the quasi-identifier data indistinguishable. Attackers can only associate the identifier attributes through the quasi-identifier attributes with a probability of 1/*k* at most. Thus K-anonymity algorithms can help achieve the privacy protection of data. Although K-anonymity can help prevent identity disclosure, it cannot guarantee the protection of sensitive attributes. Thus the algorithms may face the challenge of semantic attacks, such as skewness attacks and similarity attacks. 

Currently, several anonymization techniques, such as *l*-diversity, (*α*, *k*)-anonymity, *ε*-differential privacy, and *t*-closeness, can defend against skewness or similarity attacks. The *l*-diversity model based on the *k*-anonymity model can defend against these attacks by guaranteeing that each equivalence class’s sensitive attribute has at least *l* values. However, when the sensitive attribute values in the same equivalence class are skewed, or the sensitive attribute values belong to the same class, the privacy model is still vulnerable to skewness and similarity attacks, and it can lead to privacy disclosure. The (*α*, *k*)-anonymity model can defend against attacks by limiting the frequency of each sensitive attribute value in the equivalence class. It can help alleviate the skewness attacks to a certain extent, but it fails to guarantee the values of sensitive attributes do not belong to the same category, and it is vulnerable to similarity attacks. The *ε*-differential privacy model defends against skewness or similarity attacks by adding noise, but results in the decline of data availability. It is only practical on large datasets and is not suitable for the publishing of small datasets. The *t*-closeness model can defend against skewness or similarity attacks by measuring the distribution distance of sensitive values between equivalence classes and the dataset. It can guarantee that the distribution of both sensitive data in each equivalence class and the entire dataset does not exceed the threshold *t*. This privacy model restricts the relation between the global distribution of quasi-identifier attributes and sensitive attributes. It also weakens the relation between quasi-identifier attributes and specific sensitive data, and reduces the possibility that attackers launch skewness and similarity attacks by the distribution of sensitive data. In short, the *t*-closeness model considers the distribution of sensitive attributes rather than sensitive attribute values. Therefore, it is more secure and practical than the aforementioned privacy models.

The *t*-closeness model has the strictest privacy guarantee among the various privacy models. Moreover, the clustering algorithm can quickly aggregate multiple similar quasi-identifier data, which will help improve the data availability and reduce the algorithm’s execution time [9]. It is necessary to investigate a K-anonymity algorithm based on *t*-closeness and clustering to defend against skewness and similarity attacks. But there still remain the following problems: (1) The current clustering methods for multi-dimensional data have poor clustering quality and low clustering accuracy. It leads to the existence of multiple similar sensitive data in different clusters, and there are more similar sensitive data in the equivalence class after data anonymization. (2) The equivalence class partition methods based on the *t*-closeness model cannot reflect the diversity of sensitive data for each equivalence class, which will result in the existence of similar sensitive data in some equivalence classes. Note that they cannot defend against skewness and similarity attacks at the same time.

In view of the above problems, we study the clustering of the anonymous data. Specifically, we study the clustering of anonymous multi-dimensional sensitive data and improve the clustering algorithm of multi-dimensional sensitive data, as well as the equivalence class partition method based on the *t*-closeness model. Moreover, we propose a K-anonymity privacy protection algorithm for multi-dimensional data against skewness and similarity attacks (KAPP). KAPP effectively defends skewness and similarity attacks and improves the anonymity of multi-dimensional sensitive data. The contributions of the paper are as follows:(1)We propose a multi-dimensional sensitive data clustering algorithm based on improved African vultures optimization. More specifically, we introduce chaotic tent mapping to improve the initialization of the cluster centers and propose a clustering fitness function for multi-dimensional sensitive data based on the traditional African vultures optimization method. We also improve the global search stage and local search stage as well as optimize the selection of cluster centers.(2)We propose an equivalence class partition and generalization method based on the measurement of sensitive data’s distribution difference value. That is to say, we propose the calculation equation of the distribution difference value of sensitive data. We also merge the equivalence classes with larger difference values and increase the generalization processing of sensitive data with the most significant weight.

The remainder of the paper is organized as follows. Section 2 introduces some general concepts. Section 3 details the current protection methods against skewness and similarity attacks. Section 4 details the principles and processes of the algorithm. Section 5 describes the implementation process and pseudo-code of the algorithm. Section 6 analyzes the security of the algorithm. Section 7 conducts experimental simulations to compare and analyze algorithms. Section 8 discusses our contribution and future direction, and Section 9 presents the conclusions.

## 2. Background

In this section, we briefly introduce some general concepts and definitions used in this paper.

**Definition** **1.***The identifier attributes can directly determine personal identities, such as name and identity card.* (identifier attributes).

**Definition** **2.***The quasi-identifier attributes can indirectly determine personal identity by associating multiple attributes, such as age and zip code. A quasi-identifier attribute value is a quasi-identifier data.* (quasi-identifier attributes). 

**Definition** **3.***The sensitive attributes are relevant to personal privacy information, such as disease names and medical costs. A sensitive attribute value is a sensitive data.* (sensitive attributes).

**Definition** **4.***The multi-dimensional data can be presented in a data table in which each record(row) corresponds to one person and each column to a specific attribute. Table 1 shows an example of multi-dimensional data, where each record represents the multi-dimensional data of a person.* (multi-dimensional data). 

**Definition** **5.***An equivalence class is a set of anonymized records with the same values for all the quasi-identifier attributes, i.e., all the records in each equivalence class are indistinguishable in terms of their quasi-identifier attributes.* (equivalence class).

**Definition** **6.***If the proportion of the same sensitive data in the same equivalence class is greater than the threshold τ %, the attacker can predict the sensitive value of an individual with high probability and thus launch the skewness attack [10]. Table 2 shows an anonymized skewness attack version of Table 1, which means the multi-dimensional data in Table 2 are vulnerable to skewness attacks. Where “***” represents that data are suppressed. In Table 2, all multi-dimensional data within each equivalence class have the same quasi-identifier data for the zip code and age attributes. However, in the first equivalence class, the frequency of sensitive data in disease attributes of pneumonia is large. In this way, an attacker can exactly tell the disease of the patients in the first equivalence class if he or she knows their personal information in the first equivalence class.* (skewness attack).

**Definition** **7.***If the sensitive data in the same equivalence class with semantic similarity (i.e., the sensitive data in the same equivalence class belongs to the same category), the attacker can infer the category of an individual with high probability [11]. Table 3 shows another anonymized version of Table 1, i.e., a similarity attack version of Table 1. Similarly, the data in Table 3 are vulnerable to similarity attacks. In Table 3, all the sensitive data in the first equivalence class’s disease attribute are different, but they all belong to cancer. In this way, if an attacker knows the personal information of a patient in the first equivalence class, the attacker will know that the patient has cancer, although he or she cannot determine what type of cancer the patient has.* (similarity attack).

## 3. Related Work

K-anonymity can effectively prevent privacy disclosure and is well-known as an effective anonymity method. Although K-anonymity can avoid identity disclosure, there may exist attacks such as skewness and similarity, which make it unavailable for sensitive attribute protection. Therefore, many privacy protection methods have been proposed, such as those based on slicing [12] and Bucketization [13]. Since the clustering algorithm can quickly aggregate multiple similar quasi-identifier data, which can improve the availability of data and reduce the execution time of the algorithm. Therefore, some scholars propose clustering-based multi-dimensional K-anonymity privacy protection algorithms. For example, Piao et al. [14] propose a clustering-based privacy-preserving anonymity approach (CPPA). CPPA can achieve the privacy protection of multi-dimensional data through the K-medoid clustering and generalization algorithm. However, every cluster center selected by this clustering method must be a sample point, which will cause a decline in the clustering quality. Thaeter et al. [15] propose a scalable K-anonymous microaggregation (SKAM), which can achieve records clustering based on quasi-identifier data and can generalize quasi-identifier data by calculating each cluster center. Unfortunately, the clustering quality of this clustering algorithm is related to dimensions, and may be limited to high-dimensional datasets. Yan et al. [16] propose a weighted K-member clustering algorithm, which can realize K-anonymity for records. The algorithm uses a weighting stage and a series of weighting indicators to evaluate the outliers of records, which is convenient for screening outliers so as to maximize the availability of anonymous data. However, the clustering effect of multi-dimensional data in the above literature [14,15,16] depends on the random selection of the initial clustering center, and the clustering results are unstable. Moreover, semantic attacks such as skewness attacks and similarity attacks in the anonymity process are not considered in the aforementioned literature, which brings certain security risks.

Therefore, some scholars combine the K-anonymity algorithm with various privacy models and focus on the K-anonymity privacy protection algorithm against skewness and similarity attacks. We classify skewness attacks and similarity attacks protection schemes based on four classic privacy models, i.e., *l*-diversity, (*α*, *k*)-anonymity, *ε*-differential privacy model, and *t*-closeness, as shown in Table 4. In terms of the *l*-diversity model, Zhang et al. [17] improve the selection of the initial clustering center of the K-means clustering algorithm and cluster the sensitive attributes by calculating the sensitivity factors of sensitive attributes. It can ensure that records with similar sensitivity are partitioned into one equivalence class, and the equivalence class also satisfies *l*-diversity. The limitation of the algorithm is that it does not consider outliers, and thus will cause the degradation in clustering quality. Ren et al. [18] use anonymous vertexes and edges, as well as the influence matrix based on background knowledge, to achieve the sensitive attribute diversity and privacy protection of individuals. This algorithm catches the *k*-isomorphism graph, and generalizes the *k*-isomorphism graph vertex group about identifier attributes and sensitive attributes. Then, the algorithm generalizes the edge group about identifier attributes and sensitive attributes. Parameshwarappa et al. [19] propose a novel multi-level clustering method, which uses a non-metric weighted distance measure to improve the clustering quality. This algorithm partitions equivalence classes by multi-level clustering and requires that each equivalence class should satisfy the *l*-diversity. However, the above literatures [17,18,19] fail to guarantee that the sensitive data in each equivalence class is evenly distributed and do not belong to the same category.

In terms of the (*α*, *k*)-anonymity model, Wang et al. [20] quantify the privacy requirements of sensitive values and sensitive groups, and partition sensitive groups by agglomerative hierarchical clustering. Then, this algorithm uses the designed global search and local search clustering algorithms to partition equivalence classes and achieve data anonymity through generalization. The algorithm requires that the frequency of sensitive data in the same equivalent class with high sensitivity is less than the threshold, but it cannot guarantee the security of sensitive data with low sensitivity. Onesimu et al. [21] and Dosselmann et al. [22] generate equivalence classes through a bottom-up clustering algorithm and an improved Mondrian algorithm, respectively. Then, they generalize each equivalence class to achieve data anonymity. These algorithms limit the frequency of each sensitive data in all equivalence classes, but they can only ensure that sensitive data are different and cannot guarantee that sensitive data do not belong to the same class.

In terms of the *ε*-differential privacy model, Raffael et al. [23] propose a data anonymization algorithm that provides guarantees for k-anonymity and differential privacy. This algorithm uses attribute taxonomies together with a randomization approach and is implemented via sampling to meet differential privacy. More specifically, the search strategy employs a (randomized) best-first search through the generalization hierarchies by using a score calculated based on the given data quality metrics (i.e., information loss, discernibility, and group size). This will help release a randomized version of a given dataset. Xu et al. [24] propose a differentially private algorithm for high-dimensional data release through random projection to maximize utility, while guaranteeing privacy. More specifically, this algorithm projects a *d*-dimensional vector representation of a user’s feature attributes into a lower *k*-dimensional space by first applying a random projection, and then adding Gaussian noise to each resulting vector. However, this algorithm uses randomness to establish the projection matrix, which will cause a large amount of noise in each calculation, such that its generated synthetic data sets are hampered by the unstable utility. Tsou et al. [25] propose the (*k*, *ε*, *δ*)-anonymization synthetic data set generation mechanism before releasing the data sets to protect data privacy. This algorithm rationally replaces high-dimensional datasets with lower-dimensional datasets by using principle component analysis, and then it introduces and modifies a *k*-anonymity clustering algorithm based on the KD-Tree data structure. Moreover, this algorithm uses a random sampling procedure to generate the synthetic datasets. Li et al. [26] propose two different privacy-preserving data publishing methods. One method generates a lattice including all the possible generalization results of the input dataset with a given hierarchy, and then uses the exponential mechanism to output a specific generalization according to the utility. This method adds noise to the mapping function, and involves sampling, suppression and generalization selection. However, the sampling and suppression will only make a proportion of the data being processed. The other method clusters the data based on the number of occurrences, and then adds Laplacian noise to each cluster. This method directly adds noise to the data, resulting in a shorter runtime but introducing more noise, which has an impact on the performance. Thus, the above literatures [23,24,25,26] usually needs to add randomization to the shared data, resulting in the decline of data availability. Therefore, they are suitable for large datasets but difficult to be applied to a small dataset.

In terms of the *t*-closeness model, Wang et al. [27] propose a privacy-preserving algorithm for multiple sensitive attributes (PAMS). PAMS uses principal component analysis to reduce the multiple sensitive attributes to one-dimensional data space, sorts the new data in ascending order and partitions them into different groups. Then, PAMS selects data from each group to generate equivalence classes through the fuzzy c-means clustering method. This algorithm satisfies the *t*-closeness anonymous model, but it requires feature dimensionality reduction, resulting in a decline in the clustering quality of multi-dimensional data. Sei et al. [28] propose a privacy model containing an anonymization algorithm. In order to satisfy *t*-closeness, the algorithm changes the original records with a fixed probability and adds some completely random records. Therefore, the reconstructed records are significantly affected by these random records, and the utility of the data is reduced. Fathalizadeh et al. [29] generalize the sensitive data and use the shortest path algorithm to find *k* records with similar sensitive data to partition the equivalence classes, then can realize data anonymity by generalizing the quasi-identifier data. However, it causes a heavy cost resulting from the rearrangement of records that satisfy the equivalence class partitioning conditions after the creation of each cluster. Langari et al. [30] propose a combined anonymizing algorithm based on K-member fuzzy clustering and a firefly algorithm for anonymized database protection. This algorithm uses a modified K-member version of fuzzy c-means to create balanced clusters with at least *K* members in each cluster. Then, it uses the firefly algorithm to optimize the primary clusters and anonymize the network graph and data. Ganarde et al. [31] propose a novel anonymizing method based on multiple-graph-properties-based clustering. This method proposed a data normalization algorithm to preprocess and enhance the quality of raw data. Then, it divides the data into different clusters using multiple graph properties to satisfy the k-anonymization. However, the above literatures [27,28,29,30,31] usually cannot guarantee that the distribution of sensitive attributes of the equivalence class is skewed.

In summary, the current methods against skewness and similarity attacks have certain limitations. More specifically, the clustering quality for multi-dimensional data of the traditional clustering algorithm is not ideal, and the equivalence class partition is still prone to the same or similarly sensitive data, which is unavailable for defending against both skewness and similarity attacks.

## 4. Algorithm Principle

Some important notation in this article are described in Table 5. The implementation scheme is as follows.

Personal multi-dimensional data are required to be collected and published in practical applications of scientific research and policy-making, e.g., disease research and epidemic prevention, which need to be processed anonymously. Therefore, we propose the K-anonymity algorithm named KAPP, which is shown in Figure 1. First, KAPP preprocesses the multi-dimensional data table to be published. Secondly, KAPP improves the African vultures optimization method. It clusters multi-dimensional data according to the sensitive data distance and selects the data with the smallest quasi-identifier distance from each cluster to form the initial equivalence classes. Then, KAPP optimizes the equivalence classes with large difference values by calculating the distribution difference value between each equivalence class and the data table. Finally, KAPP generalized quasi-identifier and sensitive data with the highest sensitivity weight to achieve anonymous publication of data tables. 

### 4.1. Multi-Dimensional Data Preprocessing

The multi-dimensional data consist of identifier data, quasi-identifier data of the numerical type and categorical type, and sensitive data of the numerical type and categorical type. Note that all the multi-dimensional data should be preprocessed before sharing. Namely, KAPP deletes all identifier data, which will prevent attackers from directly associating personal information with identifier data to obtain sensitive data. Since the multi-dimensional data are divided into numerical and categorical types, it is necessary to cluster all multi-dimensional data according to numerical and categorical data. Therefore, the categorical data is transferred to a specific value. After clustering, it can be restored to the categorical data before the transformation.

### 4.2. Multi-Dimensional Data Clustering Based on Improved African Vultures Optimization

If the multi-dimensional data is partitioned into clusters and equivalence classes directly according to the pre-processed quasi-identifier data, it is easy to suffer skewness and similarity attacks. Therefore, KAPP considers clustering multi-dimensional data based on sensitive data. At the same time, the fuzzy C-means clustering algorithm can realize a flexible partition of sample points by calculating the membership degree of each sample point to all cluster centers [32]. Compared with the hard partition of other clusters, this clustering algorithm is more suitable for multi-dimensional data clustering. However, it is easy to fall into the optimal local solution in the process of solving, resulting in its clustering effect could be worse. Considering that the African vulture optimization algorithm (AVOA) divides multiple stages in solving the optimal solution and simulates the vulture’s predatory behavior to avoid local optimization [33,34]. It also has a short optimization execution time and can provide optimal solutions for various dimensions. Therefore, based on the fuzzy C-means clustering algorithm, KAPP introduces the optimal solution-searching method of AVOA. However, the initial clustering center of the fuzzy C-means clustering algorithm is randomly generated, which leads to a long convergence time in the early stage of the algorithm and an unstable clustering effect. In addition, AVOA only considers the current optimal solution to update the current solution in the global search phase, which may lead to a poor effect of the updated solution and slow convergence speed in the medium-term of the algorithm. At the same time, in the local search phase of AVOA, different optimal cluster center solutions have the same weight for updating the current solution. It cannot adjust the update effect of the optimal solution and the suboptimal solution on the current solution, resulting in the algorithm not reaching the convergence state in a short time. Therefore, we improve the cluster center’s initialization, the fitness value calculation method, and the solution update strategy in the global and local search stages. Finally, we propose a multi-dimensional data clustering algorithm based on improved African vultures optimization that can cluster multi-dimensional data based on sensitive data. The clustering process is as follows:

#### 4.2.1. Solution Initialization Based on Chaotic Mapping

KAPP initializes the solution by combining numerically sensitive data and numerically transformed categorically sensitive data. Namely, the dimension is determined according to the number of sensitive attributes in multi-dimensional data and constructs the search space for solutions. But the long convergence time caused by the random generation of the initial cluster center will make the clustering effect unstable. Considering that chaotic tent mapping can generate random sequences with pseudo randomness and distribution uniformity [35], and the distribution of the generated data can be more evenly. Therefore, KAPP combines chaotic tent mapping to randomly generate solution *x* composed of *k* cluster centers to cover the entire solution space. It improves the global search ability of the algorithm and improves the accuracy of multi-dimensional data clustering based on sensitive data. Namely, it generates a sensitive data of a single cluster center in solution *x* through Equation (1) and executes *sas* repeatedly to obtain a single cluster center $x(it)=\{csa_{1},\cdots ,csa_{si},\cdots ,csa_{sas}\}$.
(1)$$csa_{si}=rsa_{si}\times (upsa-dnsa)+dnsa$$
where *upsa* and *dnsa* represents the maximum and minimum sensitive data of the current sensitive attribute in the dataset. Note that $rsa_{si}$ represents the *si*-th random number of the chaotic tent mapping, which can be expressed as:(2)$$rsa_{si}=\left\{\begin{array}{ll} rand_{0}^{1}, & si=0\\ rsa_{si-1}\times 2, & 0\leq rsa_{si-1}<0.5\\ (1-rsa_{si-1})\times 2, & 0.5\leq rsa_{si-1}\leq 1 \end{array}\right.$$
where $rand_{l}^{r}$ represents the random number between *l* and *r*. The process repeats the cluster center selection for *k* times to obtain an initial solution, and generates *m* initial solutions.

#### 4.2.2. Adaptation Calculation and Solution Selection Improvement

KAPP takes the membership of multi-dimensional data belonging to different cluster centers as the optimization objective and proposes the fitness calculation function of the solution.
(3)$$f_{zi}=\sum_{q=1}^{k}\sum_{j=1}^{n}{\left(u_{j}^{q}\right)}^{e}\times Dist(r_{j},c_{q})$$
where $f_{zi}$ represents the fitness value of the *zi-*th solution, *e* represents the fuzzy weight, $u_{j}^{q}$ represents the membership of the *j-*th multi-dimensional data in the *q-*th cluster, which can be expressed as:(4)$$u_{j}^{q}=\frac{1}{Dist{\left(r_{j},c_{q}\right)}^{2}}/\sum_{l=1}^{k}\frac{1}{Dist{\left(r_{j},c_{l}\right)}^{2}}$$
where $r_{j}$ represents the *j-*th multi-dimensional data, $c_{q}$ represents the multi-dimensional data of the *q-*th cluster center, $c_{l}$ represents the multi-dimensional data of the *l-*th cluster center. $Dist(r_{j},c_{q})$ represents the distance between $r_{j}$ and $c_{q}$. $Dist(r_{j},c_{l})$ represents the distance between $r_{j}$ and $c_{l}$, which can be expressed as:(5)$$Dist(r_{j},c_{q})=\sum_{l=1}^{a}\frac{\left|Num_{rj}^{l}-Num_{cq}^{l}\right|}{n}+\sum_{l=1}^{b}\frac{h(Cate_{rj}^{l},Cate_{cq}^{l})}{h(H)}$$
where $Num_{j}^{l}$ represents the *l-*th numerical data in $r_{j}$, $Num_{j}^{l}$ represents the *l*-th numerical data in $c_{q}$, $h(Cate_{rj}^{l},Cate_{cq}^{l})$ represents the number of leaf node in the parent node of the *l*-th categorical data in $r_{j}$ and $c_{q}$ [36], h(H) represents the total number of sensitive data of the current categorical attributes.

KAPP calculates the fitness values of all current solutions and selects the two solutions with the lowest fitness values as the optimal solution $BV_{1}(it)$ and suboptimal solution $BV_{2}(it)$. Then, it randomly selects a solution from $BV_{1}(it)$ and $BV_{2}(it)$ as the best solution *BV*(*it*) by roulette. Where *it* represents the current number of iterations.

#### 4.2.3. Solution Update Based on Improved African Vultures Optimization

According to the current solution, KAPP introduces and improves the African vulture algorithm to update the solution. We compare each solution to a vulture. We simulate the behavior of vultures looking for food and calculate the starvation rate *HR* of each solution by Equation (6). If the starvation rate of the solutions is high, they have enough energy to go a long distance and look for the optimal solution. Otherwise, they look for the optimal solution in a nearby area.
(6)$$HR=rand_{-2}^{2}\times (\sin^{op}(\frac{\pi }{2}\times \frac{it}{Maxit})+\cos(\frac{\pi }{2}\times \frac{it}{Maxit})-1)+rand_{-3}^{3}\times (1-\frac{it}{Maxit})$$
where *op* represents the probability parameter of entering the global search stage. The specific updates are as follows:(a)When HR≥1, KAPP is in the global search stage. Since AVOA only considers the best solution *BV*(*it*) to update the current solution, the solution search space area is too large, which makes the algorithm’s medium-term convergence slow. Therefore, we record the historical optimal solution of each solution at the global search node. Then, we use the current solution’s historical optimal solution to update it to ensure that the solution is not too bad and improve the early convergence speed. Namely, the solution must be updated with the improved Equation (7). If the probability parameter is $P_{1}<rand_{0}^{1}$, it is necessary to quickly determine the search range of the optimal solution by simulating vultures searching for food in different spatial areas by Equation (7). Otherwise, it is necessary to further narrow the search range of the optimal solution by simulating vultures searching for food in the area around the random distance of the optimal solution by Equation (7).
(7)$$x(it+1)=\left\{\begin{array}{ll} BV(it)-|rand_{0}^{2}\times (BV(it)+H(it))\times \frac{1}{2}-x(it)|\times HR, & P_{1}\geq rand_{0}^{1}\\ rand_{0}^{1}\times ((upsa-dnsa)\times rand_{0}^{1}+dnsa)+BV(it)-HR, & P_{1}<rand_{0}^{1} \end{array}\right.$$
where H(it) represents the historical optimal solution of the current solution.
(b)When 1≥HR≥0.5, KAPP is in the first stage of local search. Namely, the solution must be updated with the improved Equation (8). If the probability parameter is $P_{2}<rand_{0}^{1}$, it is necessary to quickly approach the local range of the optimal solution by simulating vultures rotating and flying close to food by Equation (8). Otherwise, it is necessary to determine the global range of the optimal solution by simulating vultures competing for food by Equation (8).
(8)$$x(it+1)=\left\{\begin{array}{ll} x(it)-BV(it)+(HR+rand_{0}^{1})\times |rand_{0}^{2}\times BV(it)-x(it)|, & P_{2}\geq rand_{0}^{1}\\ BV(it)\times (1-(\frac{rand_{0}^{1}\times x(it)}{2\times \pi })\times \cos(x(it))-(\frac{rand_{0}^{1}\times x(it)}{2\times \pi })\times \sin(x(it))), & P_{2}<rand_{0}^{1} \end{array}\right.$$

(c)When HR<0.5, KAPP is in the second stage of local search. Since the weight of $BV_{1}(it)$ and $BV_{2}(it)$ to the updated solution is the same in AVOA, the algorithm cannot adjust the updated impact of the optimal solution and the suboptimal solution to the current solution, resulting in a poor convergence effect of fitness value. Therefore, we introduce weights $\omega_{1}$ and $\omega_{2}$ to control the updated impact of $BV_{1}(it)$ and $BV_{2}(it)$ on the current solution. It updates the solution by adjusting the most appropriate weights to improve later local search capabilities and cluster accuracy. Namely, it is necessary to update the solution with the improved Equation (9). If the probability parameter is $P_{3}\geq rand_{0}^{1}$, it is necessary to gradually approach the optimal solution location by simulating the massive competition of vultures for food by Equation (9). Otherwise, it is necessary to accurately capture the optimal cluster center position by simulating the fierce competition of vultures for food by Equation (9).
(9)$$x(it+1)=\left\{\begin{array}{cc} \begin{array}{c} \omega_{1}\times BV_{1}(it)\times (1-\frac{x(it)\times HR}{BV_{1}(it)-x{\left(it\right)}^{2}})\\ +\omega_{2}\times BV_{2}(it)\times (1-\frac{x(it)\times HR}{BV_{2}(it)-x{\left(it\right)}^{2}}), \end{array} & P_{3}\geq rand_{0}^{1}\\ BV(it)-|BV(it)-x(it)|\times HR\times Levy, & P_{3}<rand_{0}^{1} \end{array}\right.$$
where $\omega_{1}$ represents the weight parameter of $BV_{1}(it)$, $\omega_{2}$ represents the weight parameter of $BV_{2}(it)$, |BV(it)−x(it)| represents the distance between the current solution and the optimal solution, *Levy* represents the linear feedback coefficient.

If KAPP completes the calculation and update of *K* cluster centers of each solution in the current iteration, it will update the membership between each multi-dimensional data of all solutions and each cluster center by Equation (4). Then, it updates the fitness values of all solutions by Equation (3) and starts the following iteration process to cluster all multi-dimensional data by sensitive data until the iterative calculation is completed.

### 4.3. Equivalence Class Division Based on Sensitive Data Distribution Difference Measurement

According to the multi-dimensional data clustering results of sensitive data, the distance increase of sensitive data of the same equivalence class can effectively defend against skewness and similar attacks. Therefore, in KAPP, we propose an equivalence class partition algorithm based on the measurement of the distribution difference value of sensitive data. The algorithm builds equivalence classes, including initialization and optimization of equivalence classes, to obtain equivalence classes that can defend against skewness and similarity attacks.

#### 4.3.1. Equivalence Class Initialization

Since the equivalence class partition affects the information loss rate of anonymous data and affects the availability of shared data. Therefore, based on the clustering results, KAPP selects the multi-dimensional data with the most similar quasi-identifier data from each cluster by Equation (5) to form an equivalence class. It can complete the initialization of *n*/*k* equivalence classes, making the sensitive data of multi-dimensional data in the same equivalence class different. However, the quasi-identifier data is similar. It can reduce the information loss rate.

#### 4.3.2. Measurement and Calculation of Sensitive Data Distribution Differences and Optimization of Equivalence Classes

After the initialization of the equivalence class, all data has been partitioned into *n*/*k* equivalence classes. However, considering the different numbers of multi-dimensional data in each cluster, some with similar sensitive data are still partitioned into the same equivalence class at the later stage of equivalence class initialization, resulting in reduced diversity. It is also considered that sensitive data with high sensitivity should be protected first. Therefore, KAPP allocates weights and calculates distribution variance values for all sensitive data in multi-dimensional data of each equivalence class. Then, KAPP uses distribution difference values for numerical and categorical data in sensitive data and optimizes the equivalence classes. The specific equivalence optimization process is as follows.

For numerical data, all sensitive data in multi-dimensional data are partitioned into $N_{av}$ levels. The interval range size of each level is $(av_{\max}-av_{\min})/N_{av}$, and corresponding weights $\varphi_{y}^{h}$ are set for each level. Where $av_{\max}$ represents the maximum sensitive data of the current level, $av_{\min}$ represents the minimum sensitive data of the current level. Combining the weight of sensitive data and the weight of its attribute, KAPP calculates the absolute value of the difference between *k* sensitive data of the equivalence class and the average sensitive data. Namely, it calculates the distribution difference value of each numerical sensitive attribute of any equivalence class by the following Equation (10):(10)$$SADN_{z}^{h}=\theta_{h}\times \frac{1}{k}\times \sum_{y=1}^{k}\left|Np_{y}^{h}\times \varphi_{y}^{h}-Nave_{h}\times \varphi_{h}^{ave}\right|$$
where $\theta_{h}$ represents the weight value of the *h*-th sensitive attribute. $Np_{y}^{h}$ represents the *y*-th sensitive data of the *h*-th sensitive attribute. $\varphi_{y}^{h}$ represents the weight value of $Np_{y}^{h}$. $Nave_{h}$ represents the average value of all sensitive data for the *h*-th sensitive attribute. $\varphi_{h}^{ave}$ represents the weight value of $Nave_{h}$.

For categorical data attributes, each sensitive data serves as a leaf node of the generalization tree [36]. KAPP determines the number of levels based on the number of the leaf node of the parent node. It also determines the sensitive data in each level according to all leaf nodes under the parent node and sets corresponding weights for each level. Then, it combines the weight of sensitive data and its attribute weight and calculates the distribution difference value of a categorical sensitive data by the following Equation (11):(11)$$SDDC_{s}^{f}=\frac{1}{n}\times \sum_{\delta =1}^{n}(\frac{h(Dp_{s}^{f},Dq_{\delta }^{f})}{h(H)}\times |\varphi_{s}^{f}-\varphi_{\delta }^{f}|)$$
where $SDDC_{s}^{f}$ represents the distribution difference value of the *s*-th sensitive data of the *f*-th sensitive attribute in the equivalence class. $Dp_{s}^{f}$ represents the *s*-th sensitive data of the *f*-th sensitive attribute in the equivalence class. $Dq_{\delta }^{f}$ represents the *δ*-th sensitive data of the *f*-th sensitive attribute in the dataset. $h(Dp_{s}^{f},Dq_{\delta }^{f})$ represents the number of the leaf node in the parent node of sensitive data $Dp_{s}^{f}$ and $Dq_{\delta }^{f}$. h(H) represents the total number of sensitive data of the current sensitive attribute. $\varphi_{s}^{f}$ represents the weight value of sensitive data $Dp_{s}^{f}$. $\varphi_{\delta }^{f}$ represents the weight value of sensitive data $Dq_{\delta }^{f}$. Therefore, the equation for calculating the differential value of distribution for each sensitive attribute of any equivalence class is as follows:(12)$$SADC_{z}^{f}=\frac{1}{k}\times \theta_{f+c}\times \sum_{s=1}^{k}SDDC_{z}^{s}$$
where $\theta_{f+c}$ represents the weight value of the *f*-th categorical sensitive attribute.

KAPP calculates the distribution difference value of sensitive data of each equivalence class by the following Equation (13):(13)$$SADV_{z}=\sqrt{\sum_{h=1}^{asa}SADN_{z}^{h}+\sum_{f=1}^{bsa}SADC_{z}^{f}}$$
where *asa* represents the number of numerical sensitive attributes. *bsa* represents the number of categorical sensitive attributes.

After KAPP calculates the distribution difference value of sensitive data of all equivalence classes, if the difference value of sensitive data distribution of the equivalence class is greater than the threshold *t*, it merges the equivalence class closest to the quasi-identifier data of the equivalence class to reduce the distribution difference value. When the distribution difference values of all equivalence classes are less than *t*, the optimization of equivalence classes is completed so that the sensitive data of the same equivalence class are different and do not belong to the same category, thus effectively defending skewness and similarity attacks. Finally, all equivalence classes need to be generalized.

### 4.4. Data Generalization

After completing the equivalence class optimization, KAPP generalizes all quasi-identifier data according to the equivalence class partition results. Namely, for numerical attributes, it uniformly modifies the quasi-identifier data to the interval range of the minimum and maximum values of the equivalence class. For categorical attributes, it modifies the quasi-identifier data of the same equivalence class to the parent node through the generalization tree so that the identity information of at least *k* individuals cannot be distinguished. In addition, it generalizes sensitive data with high weight, further preventing the privacy disclosure of sensitive data with high sensitivity.

## 5. Algorithm Implementation

The data publishers execute KAPP to realize the anonymous publishing of multi-dimensional data and avoid privacy disclosure of anonymous data caused by skewness and similarity attacks. The pseudo-code of KAPP is shown in Algorithm 1. In line 1, the data publishers initialize all algorithm parameters. In line 2, the data publishers delete all identifier data and convert categorical data to the numerical value. In lines 3–7, the algorithm starts to cluster multi-dimensional data based on sensitive attributes. The data publisher calculates the *si*-th value $csa_{si}$ of the current initial cluster center through the equation. The process continues until the initialization of *m* solutions is completed. Where each solution contains *k* cluster centers, and each cluster center contains *sas* values. In line 8, *m* solutions begin to be updated iteratively. In lines 9–11, the data publisher calculates the membership and fitness values of the *m* solutions in the current iteration. In lines 12–32, the algorithm updates all cluster center positions of *m* solutions. More specifically, the data publisher calculates the starvation rate *HR* of the current solution. Then, according to the values of *HR*, $P_{1}$, $P_{2}$, $P_{3}$ and random number, the data publisher selects the corresponding equation to update the cluster center position of the current solution. The process continues until all solutions are updated in the current iteration. In line 33, *m* solutions of the current iteration are updated, and the process continues until the maximum number of iterations is reached. In lines 34–35, *m* solutions are updated. The data publisher selects the solution with the lowest fitness value as the final clustering result, and then selects the multi-dimensional data with the most similar quasi-identifier from each cluster to form *n*/*k* initial equivalence classes. In lines 36–41, the data publisher calculates the sensitive data distribution difference value between each initial equivalence class and the dataset. If the distribution difference value of the current equivalence class is greater than the threshold *t*, the data publisher merges it with the equivalence class whose quasi-identifier distance is the closest. The process continues until the distribution difference value of all equivalence classes is less than the threshold *t*. In line 42, the data publisher generalizes the quasi-identifier data of all equivalence classes and sensitive data with significant sensitive weight and outputs anonymous multi-dimensional data.
**Algorithm 1.** K-Anonymity Privacy Protection Algorithm for Multi-Dimensional Data Against Skewness and Similarity Attacks (KAPP)Input: original multi-dimensional dataOutput: anonymous multi-dimensional data1: *n* = 6000; *k* = 7; *m* = 30; *Maxit* = 200; *sas* = 6; *t* = 0.1;2: Deleting all identifier data, and converting categorical data to numerical value;3: **while** (current solution *zi* ≤ *m*) **do**
4:  **while** (current cluster center *j* ≤ *k*) **do**
5:   $csa_{si}=rsa_{si}\times (upsa-dnsa)+dnsa$; 6:   **end**
7: **end**
8: **while** (current iteration *it* ≤ *Maxit*) **do**9:   **while** (current solution *zi* ≤ *m*) **do**10:    ujq=1Dist(rj,cq)2/∑l=1k1Dist(rj,cl)2, fzi=∑q=1k∑j=1n(ujq)e×Dist(rj,cq);11:   **end**12:   **while** (current solution *zi* ≤ *m*) **do**13:     HR=rand−22×(sinop(π2×itMaxit)+cos(π2×itMaxit)−1)+rand−33×(1−itMaxit);14:     **if** (*HR* ≥ 1) **then**15:      **if** (P1 ≥ rand01) **then**16:       x(it+1)=BV(it)−|rand02×(BV(it)+H(it))×12−x(it)|×HR;17:      **else**18:       x(it+1)=rand01×((upsa−dnsa)×rand01+dnsa)+BV(it)−HR;19:      **end if**20:     **else if** (1 ≥ *HR* ≥ 0.5) **then**21:      **if** (P2 ≥ rand01) **then**22:      x(it+1)=(HR+rand01)−BV(it)+x(it)×|rand02×BV(it)−x(it)|;23:     **else**24:       x(it+1)=BV(it)×(1−(rand01×x(it)2×π)×cos(x(it))−(rand01×x(it)2×π)×sin(x(it)));25:     **end if**26:    **else**27:     **if** (P3 ≥ rand01) **then**28:        x(it+1)=ω1×BV1(it)×(1−x(it)×HRBV1(it)−x(it)2)+ω2×BV2(it)×(1−x(it)×HRBV2(it)−x(it)2);29:     **else**30:      x(it+1)=BV(it)−|BV(it)−x(it)|×HR×Levy;31:     **end if**32:    **end if**33:  **end**34: **end**35: Selecting the most similar data to form *n*/*k* equivalence classes;36: **while** (current equivalence class *v* ≤ *n*/*k*) **do**37:      SADVz=∑h=1aSADNzh+∑f=1bSADCzf;38:     **while** (SADVz>t) **then**39:     Combining the most similar equivalence classes to the quasi-identifier data;40:   **end**41: **end**42: Generalizing data and outputting anonymous multi-dimensional data;

## 6. Security Analysis

After anonymous data sharing and publishing, attackers in the network can obtain individual identifier data and quasi-identifier data in other ways. They expect to analyze the individual’s corresponding sensitive data from the shared anonymous data to obtain illegal profits. Therefore, attackers will launch different types of attacks on shared anonymous data, such as skewness attacks and similarity attacks. The anonymous algorithm KAPP proposed in this paper has high security. The specific security analysis is as follows.

**Theorem** **1.**
*Let an attacker try to discern the similarity between the sensitive attributes by skewness attacks and similarity attacks, then the distribution of sensitive attributes is t, and the probability of sensitive information disclosure is less than 1/k.*


**Proof.** KAPP requires that the distribution of sensitive attributes of each equivalence class and the dataset is less than the threshold t, so it can effectively protect sensitive attributes. Specifically, the KAPP is mainly divided into three stages: multi-dimensional data clustering, equivalence class optimization, and data generalization. In the multi-dimensional data clustering stage, KAPP clusters multi-dimensional data according to sensitive data through the multi-dimensional data clustering algorithm based on improved AVOA. It ensures that the sensitive data of multi-dimensional data in different clusters are different. In the equivalence class optimization stage, according to the clustering results, KAPP selects a piece of data with the most similar quasi-identification from different clusters to complete the initialization of the equivalence class. It ensures that the sensitive data of each multi-dimensional data in the same equivalence class is different. Therefore, KAPP reduces the probability of accurate analysis of attackers and improves the diversity and anonymity of algorithms. Moreover, we consider that the number of multi-dimensional data in each cluster is different. Some multi-dimensional data with similar sensitive data are still partitioned into the same equivalence class after the initialization. It increases the probability of accurate analysis by attackers, resulting in a decline in algorithm diversity and anonymity in the later stage. Therefore, KAPP finds the equivalence classes with low diversity by the difference value of sensitive data distribution of each equivalence class. Then, it merges the most similar equivalence classes of quasi-identifiers to reduce the distribution difference between the equivalence classes and the dataset. It ensures that the distribution of sensitive data of equivalence classes is similar to that of the dataset, thus reducing the number of individual sensitive data analyzed by attackers. Therefore, there are at least *k* multi-dimensional data in each equivalence class, and the probability of sensitive information disclosure is less than 1/*k*. □

## 7. Experimental Simulation

### 7.1. Simulation Parameters and Performance Parameters

Our experimental dataset comes from a cooperative medical information technology company in Hangzhou, China, whose total number of instances is 7527. The dataset contains the patient’s personal information and hospital visit data (14 attributes in total), including the number, age, gender, work category, visit card type, person source, visit zone, hospital name, admission time, discharge time, diagnosis name, medical cost, overall pooling cost, and payment time. Note that each patient number is unique, and is categorized as an identifier attribute. However, age, gender, work category, and person source are general information to patients, and are categorized as quasi-identifier attributes. Moreover, age is a numerical attribute, and the remaining quasi-identifier attributes are categorical attributes. Visit card type, visit zone, hospital name, admission time, discharge time, diagnosis name, medical cost, overall pooling cost, and payment time are sensitive medical information of patients, and they are categorized as sensitive attributes. Medical cost and overall pooling cost are numerical attributes, while other sensitive attributes are category attributes.

In order to verify the performance of KAPP in the test environment of Intel i5-9400F CPU 2.20 GHz, 16 G memory, and GTX1660 graphics card, this paper uses the experimental parameters shown in Table 6 and Python language to achieve the privacy protection for multi-dimensional data. The probability parameters $\left\{P_{1},P_{2},P_{3}\right\}$ and weight parameters $\left\{\omega_{1},\omega_{2}\right\}$ are determined by experiments. The fuzzy weight *e* is determined by relevant literature [32], and other parameters are consistent with AVOA [33]. We replicate skewness and similarity attacks in the experiment and attack the anonymized data. After the data anonymization, we count the equivalence classes whose proportion of the same number on sensitive attribute values or their ancestor node values is greater than the threshold τ%. Thus, the attackers can successfully analyze the privacy information of individuals from the equivalence classes. Based on the attack results, we study the impact of the maximum number of iterations on the convergence effect of the algorithm’s fitness value. Then, we study the impact of the constraint parameter *t* and the number of clusters *K* on anonymity. We use KAPP, CPPA [14], SKAM [15], and PAMS [27] to study the diversity, clustering accuracy, anonymity, and information loss rate under the different numbers of multi-dimensional data and the different numbers of sensitive attributes. The diversity *VD* reflects the distribution of sensitive data in the same equivalence class [37]. The greater the diversity of the K-anonymity algorithm, the higher the degree of privacy protection is, the more evenly the sensitive data distributed in the equivalence class becomes, and the more difficult it is for the attacker to derive the sensitive data of specific individuals. 

The equation for calculating the diversity of equivalence classes is as follows:(14)$$VD(E_{v})=(\sum_{i=1}^{q-1}\sum_{j=i+1}^{q}Dist(r_{i}^{v},r_{j}^{v}))/q$$
where $VD(E_{v})$ represents the diversity of the *v*-th equivalence class. $Dist(r_{i}^{v},r_{j}^{v})$ represents the sensitive data distance between the *i*-th and *j-*th multi-dimensional data of the *v*-th equivalence class. *q* represents the number of multi-dimensional data in the current equivalence class. The equation for calculating the diversity of an anonymous multi-dimensional data table is as follows:(15)$$VD(T)=\sum_{v=1}^{cn}VD(E_{v}))/cn$$
where VD(T) represents the average diversity of an anonymous multi-dimensional table. The clustering accuracy reflects the clustering algorithm’s division of clusters. It is defined as the correctly divided number of multi-dimensional data divided by the total number of data. The anonymity reflects the privacy protection of shared data after anonymity [38]. It is defined as the number of specific individuals’ real sensitive data that cannot be successfully analyzed by attackers divided by the total number of data under skewness and similarity attacks. The calculation equation is as follows.
(16)$$Anony=\frac{n-RecoNum_{sim}-RecoNum_{skew}}{n}$$
where $RecoNum_{sim}$ represents the total number of multi-dimensional data whose all sensitive data of a sensitive attribute in the same equivalence class belong to the same parent node. $RecoNum_{skew}$ represents the total number of multi-dimensional data whose proportion of sensitive data in the same equivalence class is greater than the threshold τ%. The information loss rate reflects the availability of shared data after anonymity. We use the de facto standard information loss metric in [27] to measure the information loss.

### 7.2. Simulation Analysis

#### 7.2.1. Influence of Parameter Selection

We analyze the impact of iteration number, probability parameter$\left\{P_{1},P_{2},P_{3}\right\}$, weight parameter$\left\{\omega_{1},\omega_{2}\right\}$, classification number *K* and constraint parameter *t* on the algorithm performance through experiments. Moreover, we take iteration number, classification number *K*, and constraint parameter *t* as examples to illustrate the impact of some parameters on algorithm clustering and equivalence class partition.

First, we select the maximum number of iterations as 200, and other parameters are shown in Table 6. Then, we analyze the influence of the algorithm iteration number on the fitness value. As shown in Figure 2, when the number of iterations is less than 150, the fitness value decreases rapidly with the number of iterations. When the number of iterations is more significant than 150, the fitness value converges to the optimal solution. The reason is that KAPP combines chaotic mapping and improves solution initialization to make the initial solution more evenly distributed. At the same time, we select weight parameters $\left\{\omega_{1},\omega_{2}\right\}$ and probability parameters $\left\{P_{1},P_{2},P_{3}\right\}$ at different stages through experiments. In the global search phase, KAPP updates the solution by combining the solution’s historical information to reduce the algorithm’s divergence in the global search phase and jump out of the optimal local solution. In the local search phase, KAPP uses weights $\omega_{1}$ and $\omega_{2}$ to control the update impact of the optimal and suboptimal solutions on the solution. It can adjust the most suitable weight to update the solution to improve the local search ability and accelerate the convergence speed in the later period. Therefore, as the number of iterations increases, the fitness value decreases rapidly and finally converges. Moreover, the maximum number of iterations *Maxit* is set to 150.

Next, we select the number of multi-dimensional data as 1000, 2000, 3000, 4000, 5000, and 6000, respectively. The number of clusters is respectively selected as 4, 5, 6, 7, and 8. Let the parameter *t* be 0.1, and other parameters are shown in Table 6. Then, we analyze the influence of cluster number *K* on anonymity. As shown in Figure 3, as the number of multi-dimensional data increases, the number of equivalence classes with similar sensitive data attributes increases. The attackers can successfully analyze the number of sensitive data of patients, so anonymity is slowly declining. As the number of clusters increases, the number of multi-dimensional data in each equivalence class starts to increase, and the proportion of sensitive attribute values and their parent node values in each equivalence class that is the same decreases. The attackers can successfully analyze that the number of patient-sensitive data decreases and the anonymity increases. However, as the number of clusters increases, the number of multi-dimensional data to be uniformly generalized increases gradually, leading to the increase of information loss rate. Therefore, anonymity and information loss rate should be comprehensively considered when selecting the number of clusters. When the number of clusters *K* is 7 and 8, there is little difference in anonymity. Considering that the increase in the number of clusters increases the information loss rate. Therefore, when the number of clusters *K* is 7, it can ensure that the algorithm has a high degree of anonymity, and the information loss rate tends to be the lowest.

Then, we select the number of multi-dimensional data as 1000, 2000, 3000, 4000, 5000, and 6000, respectively. The constraint parameter *t* is respectively selected as 0.1, 0.2, 0.3, 0.4, and 0.5, and other parameters are shown in Table 6. Then, we analyze the influence of constraint parameter *t* on anonymity. As shown in Figure 4, more equivalence classes of similar sensitive data are generated with the increase in the number of multi-dimensional data. It leads to a smaller average diversity of sensitive data in equivalence classes and more equivalence classes with the same number of sensitive attribute values and their parent node value accounting for more than the threshold τ%. The attackers successfully analyze more sensitive data of patients. Therefore, the anonymity of KAPP decreases, but the decrease is limited. With the gradual increase of the constraint threshold parameter *t*, the distribution requirements of sensitive data for equivalence class partition are reduced, and the number of equivalence classes that do not meet the constraint is reduced. Sensitive data in the same equivalence class are similar, so it is challenging to show diversity. It leads attackers to analyze the number of sensitive data of patients successfully, so its anonymity decreases. When the parameter *t* is 0.4 and 0.5, in the process of equivalence class partition, fewer equivalence classes do not meet the constraint conditions and are merged. At this time, the parameter *t* has little influence on the equivalence class partition, and its anonymity has little difference. Therefore, when the parameter *t* is 0.1, the algorithm has a high degree of anonymity.

#### 7.2.2. Performance Analysis as the Amount of Multi-Dimensional Data Changes

The number of multi-dimensional data is selected as 1000, 2000, 3000, 4000, 5000, and 6000, respectively. Let the parameter *t* be 0.1, the number of clusters is 7, and other parameters are shown in Table 6. Then, we analyze and compare the clustering accuracy, diversity, anonymity, and information loss rate of KAPP, CPPA, PAMS, and SKAM under the different numbers of multi-dimensional data. First, we analyze the clustering accuracy of algorithms with the different numbers of multi-dimensional data. As shown in Figure 5, with the increase in the number of multi-dimensional data, the clustering accuracy of PAMS, CPPA, and SKAM fluctuates wildly. In contrast, the clustering accuracy of KAPP fluctuates slightly and is significantly higher than other algorithms. The reason is that KAPP combines a multi-dimensional data clustering algorithm and generates the initial population through chaotic tent mapping. It can search for a better initial cluster center position before each clustering and combine the fuzzy C-means clustering algorithm and African vultures optimization algorithm. It can also accurately search the optimal global solution by improving the fitness calculation method and the current solution update strategy to achieve clustering according to the multi-dimensional data of sensitive data, which is more suitable for multi-dimensional data clustering. However, the clustering effect of other algorithms depends on the random selection of the initial clustering center. They tend to fall into the optimal local solution, and the clustering effect could be better. Therefore, KAPP’s clustering accuracy fluctuates less and is always higher than other algorithms.

We further analyze the algorithm diversity with different numbers of multi-dimensional data. As shown in Figure 6, with the increase in the number of multi-dimensional data, the diversity of KAPP, PAMS, CPPA, and SKAM shows a downward trend. Note that the diversity of KAPP is significantly higher than that of PAMS, CPPA, and SKAM. The reason is that KAPP completes the clustering of multi-dimensional data and initialization of equivalence classes based on sensitive data and merges equivalence classes that do not meet the distribution conditions. It makes the distribution of sensitive data in equivalence classes become more evenly, and each equivalence class’s multi-dimensional data increase in the sensitive data distance. However, other algorithms only consider quasi-identifier data rather than sensitive data. It will make the sensitive data of the same equivalence class more similar, and the sensitive data distance between multi-dimensional data is smaller. Therefore, the diversity of KAPP is always superior to other algorithms.

Then, we analyze the anonymity of algorithms with different numbers of multi-dimensional data. As shown in Figure 7, as the number of multi-dimensional data increases, the anonymity of KAPP, PAMS, CPPA, and SKAM remains unchanged. However, the anonymity of KAPP is higher than that of other algorithms. The reason is that KAPP uses a multi-dimensional data clustering algorithm based on improved African vultures optimization to cluster multi-dimensional data with high accuracy. Then, it partitions the equivalence classes according to the quasi-identifier data and merges the equivalence classes that do not meet the constraint conditions. It makes the sensitive data of the same equivalence class different and does not belong to the same category. Attackers successfully analyze less patient-sensitive data. However, the clustering effect of other algorithms is not ideal. In addition, they only consider that sensitive data of the same equivalence class is different and do not consider that sensitive data of the same class also cause privacy disclosure. Therefore, the anonymity of KAPP is significantly higher than that of other algorithms under skewness and similarity attacks.

Finally, we analyze the information loss rate of each algorithm with the different numbers of multi-dimensional data. As shown in Figure 8, with the increase in the number of multi-dimensional data, the information loss rate of KAPP, PAMS, CPPA, and SKAM shows a downward trend. Moreover, the information loss rate of KAPP is lower than that of PAMS and slightly higher than that of CPPA and SKAM. The reason is that with the increase in the number of multi-dimensional data, the number of multi-dimensional data similar to quasi-identifier data increases, and each equivalence class can find more similar multi-dimensional data, so the information loss rate appears to be declining. At the same time, in optimizing the privacy protection model, KAPP considers that similar sensitive data in the same equivalence class are vulnerable to attacks and clusters multi-dimensional data with similar sensitive data. Then, it selects the most similar multi-dimensional data of quasi-identifier data from each cluster to form an equivalence class and optimizes the equivalence class that does not meet the constraint conditions. Therefore, KAPP increases the process of protecting sensitive data and has higher requirements for classifying multi-dimensional data equivalence classes. Its information loss rate slightly increases. However, CPPA and SKAM are not considered the distance of sensitive data in the process of equivalence classification and only partition the equivalence class for quasi-identifier data, so their information loss rates are slightly lower than KAPP. PAMS partitions equivalence classes through traditional clustering, and the clustering effect is not ideal, so its information loss rate is the highest. However, the information loss rate of KAPP is similar to that of PAMS and CPPA. Although KAPP has sacrificed some information loss, it has dramatically improved its diversity and anonymity.

#### 7.2.3. Performance Analysis as the Number of Sensitive Attributes Changes

We analyze and compare the diversity, anonymity, and information loss rate of KAPP, CPPA, PAMS, and SKAM under the different numbers of sensitive attributes. Let the parameter *t* be 0.1, the number of clusters is 7, the number of multi-dimensional data still be 6000, and other parameters are shown in Table 6. Firstly, we analyze the algorithm diversity with different numbers of sensitive attributes. As shown in Figure 9, as the number of sensitive attributes increases, the diversity of KAPP, PAMS, CPPA, and SKAM shows a downward trend. Moreover, the KAPP’s diversity is significantly higher than that of other algorithms. The reason is that with the increase in the number of sensitive attributes, it is difficult to ensure that each dimension of sensitive attributes meets the diversity requirements. The number of equivalence classes that do not meet diversity rises. Thus diversity tends to decline. Moreover, KAPP improves AVOA and can find the optimal solution with different dimensions. At the same time, KAPP merges equivalence classes to improve diversity. However, other algorithms are inaccurate for multi-dimensional data clustering, and they balance privacy and availability by finding the optimal equivalence class. They cannot highlight the diversity of equivalence classes. Therefore, the diversity of KAPP is superior to other algorithms.

Then, we analyze the anonymity of the aforementioned algorithms with the different numbers of sensitive attributes. As shown in Figure 10, as the number of sensitive attributes increases, the anonymity of KAPP, PAMS, CPPA, and SKAM shows a downward trend. Moreover, KAPP’s diversity is significantly higher than that of other algorithms. The reason is that KAPP improves the AVOA and merges equivalence classes with small diversity. The diversity of KAPP with different numbers of sensitive attributes is always higher than that of other algorithms. The greater the diversity is, the smaller the number of patients’ sensitive data that can be successfully analyzed by attackers. Therefore, it can be concluded that the anonymity of KAPP is superior to other algorithms.

Finally, we analyze the algorithms’ information loss rate with different numbers of sensitive attributes. As shown in Figure 11, with the increase in the number of sensitive attributes, the information loss rates of KAPP, PAMS, CPPA, and SKAM all show an upward trend. The reason is that with the increasing number of sensitive attributes, it will be difficult to ensure the diversity of each equivalence class’s multi-dimensional sensitive data. Many equivalence classes do not satisfy the partition conditions, and KAPP needs to merge more equivalence classes. It leads to an increased information loss rate after the generalization of each equivalence class. Other algorithms must constantly adjust the equivalence class, and the algorithm diversity is guaranteed by sacrificing the information loss rate, resulting in a larger rate of information loss over time.

## 8. Discussion

The proposed KAPP algorithm combines the improved African vultures optimization and the proposed equivalence class partition method to anonymize the multi-dimensional dataset. The algorithm aims to improve the diversity of sensitive data in each equivalence class to defend against skewness attacks and similarity attacks, so as to prevent the attacker from re-identifying the sensitive data of individuals. More specifically, the improved African vultures optimization method first uses chaotic tent mapping to initialize cluster centers and achieve more stable clustering results. Then, it introduces the concept of membership degree in the calculation of fitness value to achieve more reasonable multi-dimensional data clustering. Finally, in the equation of updating the cluster center position, it introduces the historical optimal solution of the cluster center, and adds weight value to better update the cluster center position and improve the clustering accuracy. We consider the multi-dimensional data clustering based on sensitive data, where the multi-dimensional data in each cluster are not similar to sensitive data. The proposed equivalence class partition method first selects a multi-dimensional data with the most similar quasi-identifier data from each cluster to complete the equivalence class initialization, so as to ensure that the sensitive data in each equivalence class are as different as possible. Then, it uses the proposed equation to calculate the sensitive data’s distribution difference value of each equivalence class and dataset. It also reduces the distribution difference value by merging the equivalence classes whose distribution difference values are large until *t*-closeness is satisfied. In addition, the method introduces weight into the calculation equation of distribution difference value, so as to preferentially protect the multi-dimensional data that is more sensitive. The effectiveness evaluation of this method is based on measuring the clustering accuracy, diversity and anonymity. The experimental results show that the clustering accuracy of KAPP is high and relatively stable. KAPP first clusters multi-dimensional data based on sensitive data, and then selects multi-dimensional data from each cluster to partition equivalence classes. It improves diversity and helps defend against skewness and similarity attacks, which is superior to other similar methods.

Next, we discuss the future direction. It is worth mentioning that our proposed method is available for the privacy protection of static datasets. However, with the continuous development of cloud computing, the Internet of Things and other technologies, dynamic datasets publishing will become more common in practical applications. If the data in several consecutively published data tables changes, to avoid privacy disclosure, it is necessary to eliminate the inner link between the anonymous tables after multiple publishing as possible. In addition, the sensitive information application disclosure of data subjects can result from the applications of data analytics and machine learning methods to large distributed data archives. Data often contains sensitive identifiable information, and even if these are protected, the excessive processing capabilities of current machine learning methods might facilitate the identification of individuals, raising privacy concerns. Therefore, the future direction can be the integration of anonymous methods suitable for dynamic datasets with data analytics and machine learning methods.

## 9. Conclusions

In this paper, in order to achieve the privacy protection of multi-dimensional data, we propose a multi-dimensional data K-anonymity privacy protection algorithm (KAPP) against skewness and similarity attacks. Firstly, we preprocess the multi-dimensional data to be published. Secondly, we propose a clustering algorithm based on improved African vultures optimization, which can accurately and efficiently cluster multi-dimensional data based on sensitive attributes. More specifically, we improve the initialization, fitness calculation, and solution update strategy of the clustering center. Thirdly, we select the nearest quasi-identifier data from each cluster to form the initial equivalence class. Then, we propose the calculation equation of the distribution difference value of sensitive data, and optimize the equivalence class by calculating the distribution difference value of sensitive data. Finally, we generalize all equivalence classes to realize multi-dimensional data anonymity. Moreover, we analyze the security of KAPP against skewness and similarity attacks, as well as the influence of parameter selection on KAPP. We also analyze the clustering accuracy, diversity, anonymity, and information loss rate of KAPP. Then we compare the performance of KAPP, SKAM, PAMS, and CPPA. The experimental results show that KAPP improves clustering accuracy, diversity, and anonymity more than other similar methods under skewness and similarity attacks, and the average information loss is not much different.

In the future, we plan to study efficient anonymization algorithms for dynamic data with data analytics and machine learning methods. This may help solve the privacy disclosure problem caused by the internal relationship among anonymous tables with similar multi-dimensional data.

## Figures and Tables

**Figure 1 sensors-23-01554-f001:**
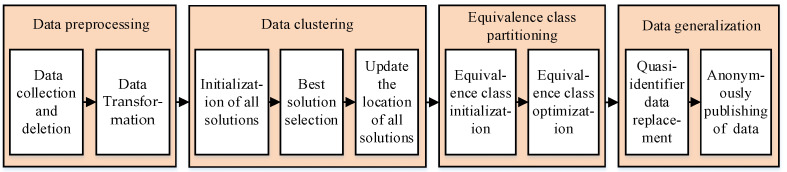
Schematic diagram of KAPP.

**Figure 2 sensors-23-01554-f002:**
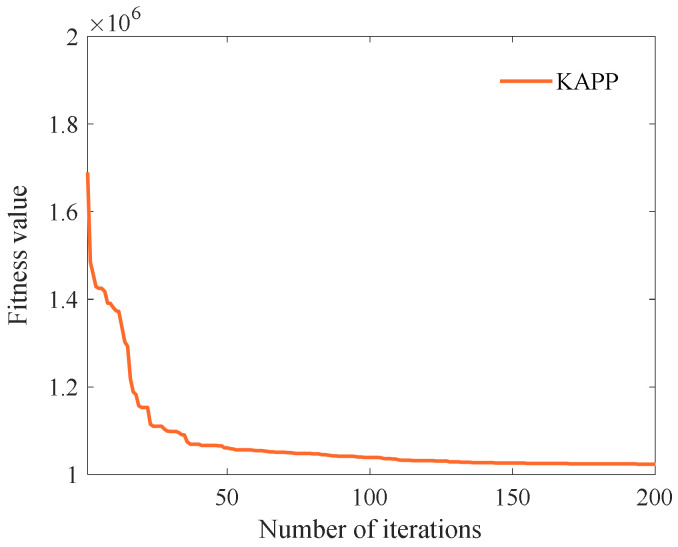
The fitness values of different iteration times.

**Figure 3 sensors-23-01554-f003:**
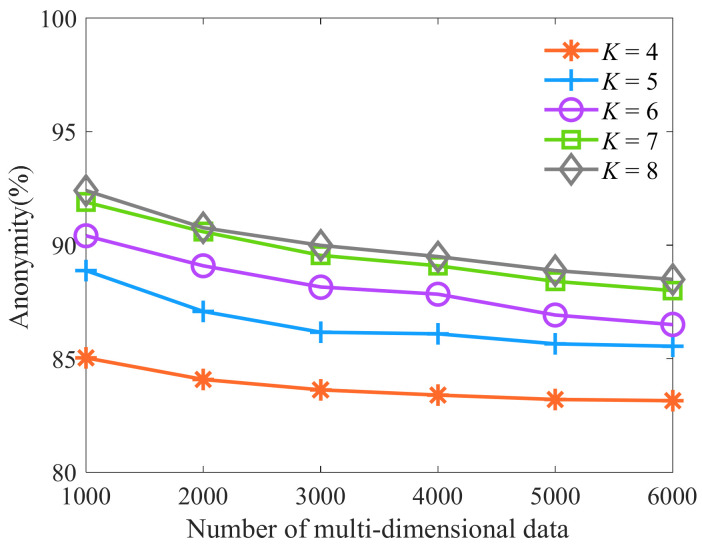
Algorithm anonymity of different clustering number *K*.

**Figure 4 sensors-23-01554-f004:**
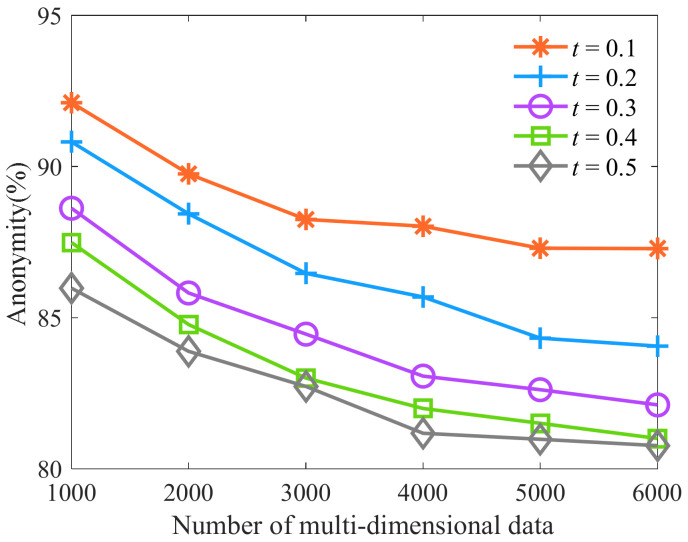
Algorithm anonymity of different constraint parameters *t*.

**Figure 5 sensors-23-01554-f005:**
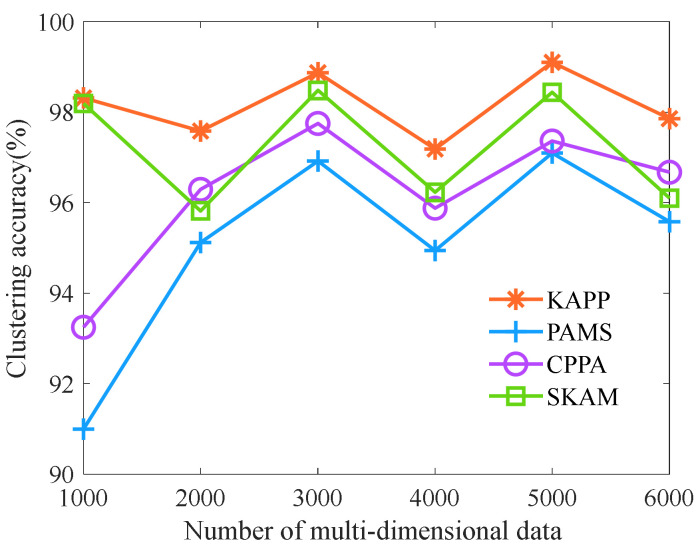
Comparison of clustering accuracy.

**Figure 6 sensors-23-01554-f006:**
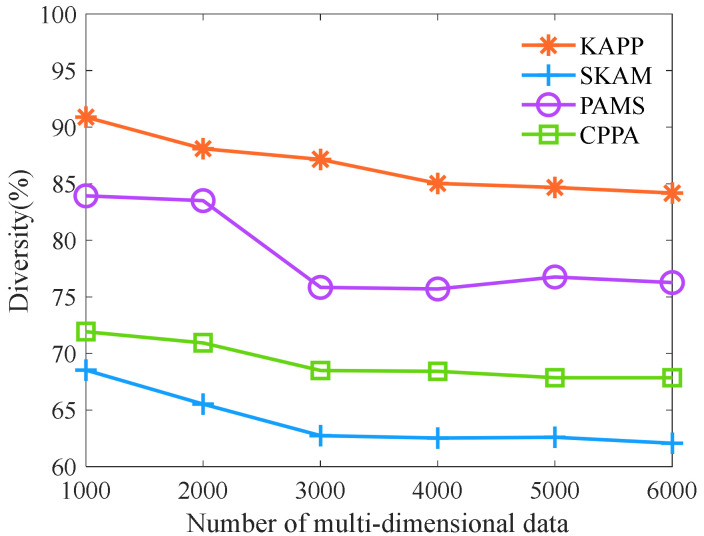
Comparison of diversity.

**Figure 7 sensors-23-01554-f007:**
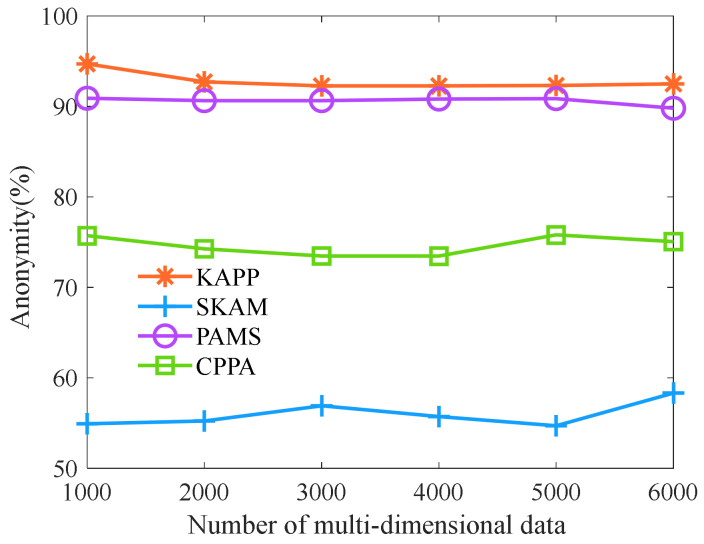
Comparison of anonymity.

**Figure 8 sensors-23-01554-f008:**
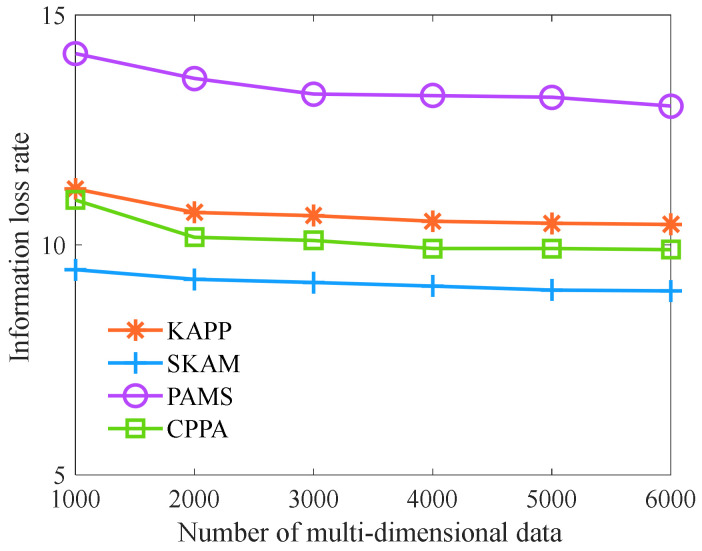
Comparison of information loss rate.

**Figure 9 sensors-23-01554-f009:**
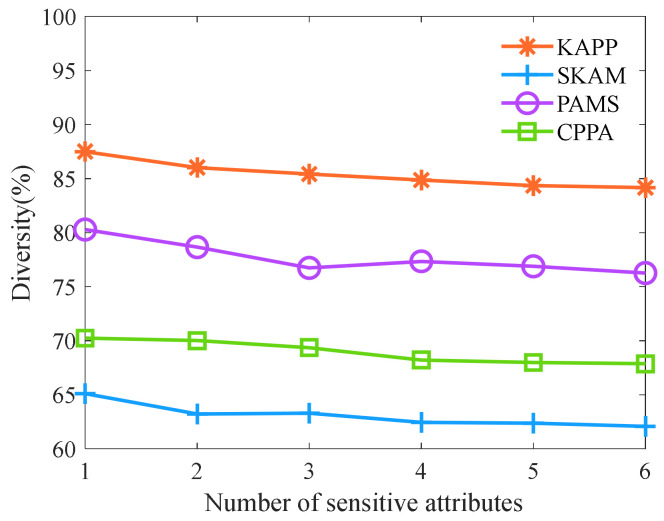
Diversity of the different numbers of sensitive attributes.

**Figure 10 sensors-23-01554-f010:**
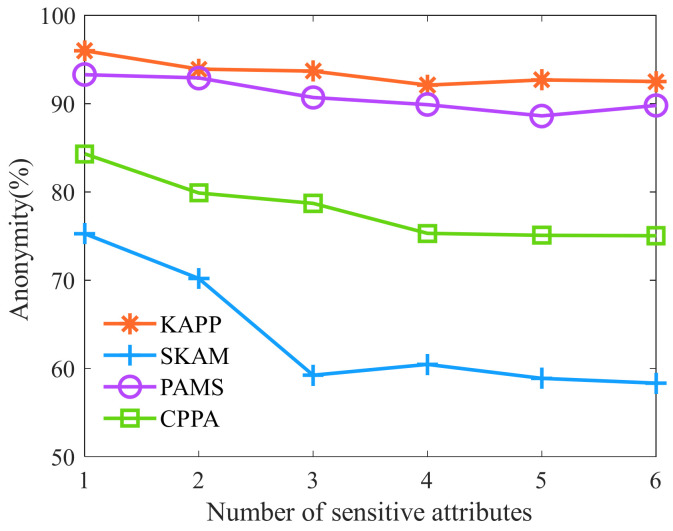
Anonymity of the different numbers of sensitive attributes.

**Figure 11 sensors-23-01554-f011:**
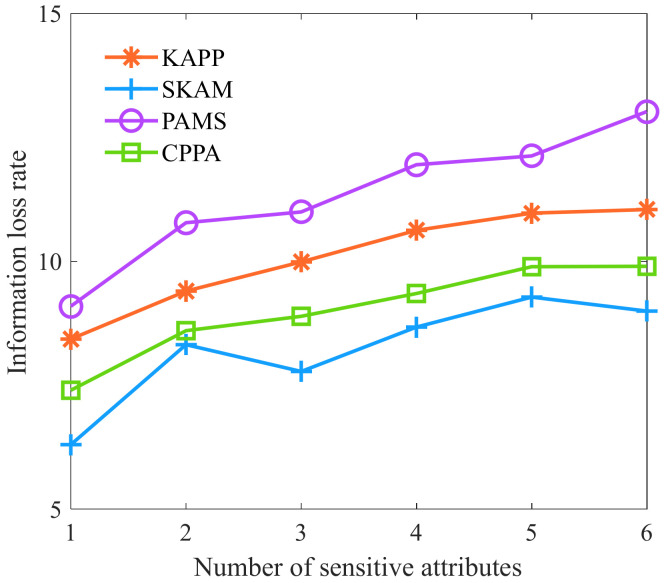
Information loss rate with different numbers of sensitive attributes.

**Table 1 sensors-23-01554-t001:** Example of personal electronic health record.

No.	Name	Age	Zip Code	Disease	Medical Cost
1	Ackerley	23	47506	Pneumonia	1000
2	Gael	26	47571	Pneumonia	1000
3	Rehor	29	47575	Breast cancer	4200
4	Jerzy	34	47603	Flu	132
5	Cade	38	47614	Colon cancer	5000
6	Finley	40	47627	Bronchitis	2000
7	Eartha	45	47709	Colitis	1500
8	Keyon	39	47714	Colon cancer	6500

**Table 2 sensors-23-01554-t002:** A skewness attack version of Table 1.

No.	Age	Zip Code	Disease	Medical Cost
1	23–29	47 ***	Pneumonia	1000
2	23–29	47 ***	Pneumonia	1000
3	23–29	47 ***	Breast cancer	4200
4	34–40	47 ***	Colon cancer	6500
5	34–40	47 ***	Bronchitis	2000
6	34–40	47 ***	Flu	132
7	38–49	47 ***	Colitis	1500
8	38–49	47 ***	Stomach cancer	8000

**Table 3 sensors-23-01554-t003:** A similarity attack version of Table 1.

No.	Age	Zip Code	Disease	Medical Cost
1	29–49	47 ***	Breast cancer	4200
2	29–49	47 ***	Stomach cancer	8000
3	29–49	47 ***	Colon cancer	6500
4	23–38	47 ***	Pneumonia	1000
5	23–38	47 ***	Colon cancer	5000
6	23–38	47 ***	Flu	132
7	26–45	47 ***	Colitis	1500
8	26–45	47 ***	Pneumonia	1000

**Table 4 sensors-23-01554-t004:** Summary of anonymization algorithms.

Literature	Privacy Model	Anonymity Technique	Methods	Utility Metrics
[17]	*l*-diversity, *k*-anonymity	suppression	k-means clustering, personalized constraints	discernibility metric, hidden ratio
[18]		suppression, generalization	anonymous vertexes and edges, influence matrix	information loss
[19]		generalization	multi-level clustering	information loss, execution time
[20]	(*α*, *k*)- anonymity	generalization	greedy clustering, personalized constraints	sensitive group recognition rate, information loss
[21]		suppression, generalization	bottom-up clustering	NCP, discernibility metric, execution time
[22]		generalization	adapted Mondrian algorithm	NCP, discernibility metric, classification metric
[23]	*ε*-differential privacy, *k*-anonymity	randomization, generalization	greedy algorithm	information loss
[24]		randomization	random projection	error rate, misclassification rate
[25]		randomization, generalization	principle component analysis, KD-Tree clustering	error rate
[26]		randomization, generalization, suppression	greedy algorithm, clustering	objective ratio, feasibility, execution time, suppression rate
[27]	*t*-closeness, *k*-anonymity	generalization	principal component analysis, fuzzy C-means clustering	information loss, execution time
[28]		randomization	similarity function	distribution distance, execution time
[29]		generalization	shortest path	information loss, NCP
[30]		pseudonymization	k-member fuzzy clustering, firefly algorithm	clustering error, information loss, execution time
[31]		edge anonymization	multiple-graph-properties-based clustering	anonymization degree, information loss, execution time

**Table 5 sensors-23-01554-t005:** Notations and definitions.

Notation	Definition
*sas*	Number of sensitive attributes
*a*	Number of numerical attributes
*b*	Number of categorical attributes
*m*	Number of solutions
*k*	Number of clusters
*n*	Total number of multi-dimensional data
*BV*(*it*)	Best solution
*HR*	Starvation rate
*Maxit*	Maximum number of iterations
$P_{1}$	Probability parameter of update mechanism in the global search phase
$P_{2}$	Probability parameter of update mechanism in the local search phase
$P_{3}$	Probability parameter of update mechanism in the local development phase
x(it+1)	Updated solution location
$SADN_{z}^{h}$	Distribution difference value of the *h*-th numerical sensitive attribute in the *z*-th equivalence class
$SADC_{z}^{f}$	Distribution difference value of the *f*-th categorical sensitive attribute in the *z*-th equivalence class
$SADV_{z}$	Distribution difference value of sensitive data of the *z*-th equivalence class
*t*	Maximum distribution difference value

**Table 6 sensors-23-01554-t006:** Simulation parameters table.

Parameter	Number	Parameter	Number
probability parameters *op*	2.5	probability parameters $p_{1}$	0.6
number of solutions *m*	30	probability parameters $p_{2}$	0.4
dimension of sensitive data *o*	6	probability parameters $p_{3}$	0.6
coefficient of linear feedback *Levy*	0.01	weight value of optimal solution $\omega_{1}$	0.6
fuzzy weight *e*	2	weight value of suboptimal solution $\omega_{2}$	0.4

## Data Availability

Not applicable.

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
