# Peer review of "K-Anonymity Privacy Protection Algorithm for Multi-Dimensional Data against Skewness and Similarity Attacks"

_sensors, 2023, doi:10.3390/s23031554_

Round 1
Reviewer 1 Report
Abstract:
First line of the abstract doesn't form a correct sentence. Rephrase it.
Abstract should clearly define the problem statement. The authors have mentioned multiple techniques to improve the clustering techniques, equivalence class partition and so on. But the main objective for doing this is not clear.
What is the role of African vulture optimization in this research can be briefed.
How the experimental results are evaluated? Performance details metrics required.
How the proposed approach is thwarting the skew and semantic attacks?
- Considering the above queries, please rewrite the abstract for better clarity.
Introduction:
The first two paragraphs are mainly focused on multi-dimensional data protection however, multi-dimensional data is not introduced. Introduce it with a sample table or records.
In sensitive attribute protection, there are single sensitive attribute protection and multiple sensitive attribute protection. Authors must clearly specify the type of problem they are addressing.
Line 62 is not true. There is a huge amount of literature study available. The authors must identify and cite literature that works on skew and similarity attacks.
Why not l-diversity, t-closeness, and their relevant study not mentioned in this section? These techniques are mainly proposed to address Skewness attacks and similarity attacks. There are much more like bucketization, and so on available.
Authors must explore the current state-of-the-art techniques and identify the research gap to propose the solution. A mere statement like "less study" is not acceptable when there is more literature already available.
Skew and Similarity attacks can be better explained with a sample table data.
Related Works:
This section should review the current state-of-the-art literature to formulate the research questions and to identify research gaps. But I feel this section just a summary or overview of different literatures. There is no real analysis found. Perform more analysis.
I suggest the authors to include a table that compares different techniques in the literature and their ways to handle different privacy attacks.
In line no. 151, the authors summarised a research paper and wrote "equivalence class is less than the threshold value α" what is α here? Author must understand the context and use appropriate words not just the words from articles.
3.1. Multi-dimensional Data Preprocessing
In this section, there is no information on the data preprocessing. Report on the preprocessing used in this work.
What is "sas" in line 240?
Check all the equations for clarity and correctness. Some of the terms used in the equations are not explained.
Algorithm1 should use the equations presented above and show the step by step process to anonymise the data. But currently the algorithm is just short of sentences which are already written. I suggest the authors, write the algorithm with less words and more mathematically. Rewrite the algorithm considering my comments.
5. Security Analysis:
This section should identify the potential challenges the proposed approach would face and how it can be tackled. It would be better if you can provide theorem and proof for various security analysis.
The citations for diversity VD and Loss metrics are missing.
In figure 3 & 4 how anonymity percentage is calculated?
Most of the privacy preserving study reports the anonymization process with respect to the change in the number of sensitive attributes. It is missing in this manuscript.
Some of the de facto standard information loss and data utility metrics are not used in this work.
Conclusion is not supported by results. No future work is available.
Some the recent references on privacy preserving studies are missing. The following may be considered.
Onesimu, J. Andrew, J. Karthikeyan, Jennifer Eunice, Marc Pomplun, and Hien Dang. "Privacy Preserving Attribute-Focused Anonymization Scheme for Healthcare Data Publishing." IEEE Access 10 (2022): 86979-86997.
Jayapradha, J., M. Prakash, Youseef Alotaibi, Osamah Ibrahim Khalaf, and Saleh Ahmed Alghamdi. "Heap Bucketization Anonymity—An Efficient Privacy-Preserving Data Publishing Model for Multiple Sensitive Attributes." IEEE Access 10 (2022): 28773-28791.
Ni, Chunchun, Li Shan Cang, Prosanta Gope, and Geyong Min. "Data anonymization evaluation for big data and IoT environment." Information Sciences 605 (2022): 381-392.
Author Response
Authors must explore the current state-of-the-art techniques and identify the research gap to propose the solution. A mere statement like "less study" is not acceptable when there is more literature already available.
Author response: Thanks for the suggestion. We have revised the aforementioned errors in the revised manuscript. In addition, we have explored and cited new relevant literature in related work, including the research that works on skewness and similarity attacks based on l-diversity, t-closeness, p-sensitivity, bucketization, slicing and clustering. The newly added literature is as follows.
- Onesimu, J.; Karthikeyan, J.; Eunice, J.; Pomplun, M.; Dang, H. Privacy preserving attribute-focused anonymization scheme for healthcare data publishing. IEEE Access. 2022, 10, 86979-86997.
- Jayapradha, J.; Prakash, M.; Alotaibi, Y.; Khalaf, O.; Alghamdi, S. Heap bucketization anonymity-an efficient privacy-preserving data publishing model for multiple sensitive attributes. IEEE Access. 2022, 10, 28773-28791.
- Parameshwarappa, P.; Chen, Z.; Koru, G. Anonymization of daily activity data by using l-diversity privacy model. ACM Transactions on Management Information Systems. 2022, 12, 1-21.
- Dosselmann, R.; Hamilton, H. Limiting sensitive values in an anonymized table while reducing information loss via p-proportion. Security and Privacy. 2021, 5, 1-21.
- Sei, Y.; Okumura, H.; Takenouchi, T.; Ohsuga, A. Anonymization of sensitive quasi-identifiers for l-diversity and t-closeness. IEEE Transactions on Dependable and Secure Computing. 2019, 16, 580-593.
- Fathalizadeh, A.; Moghtadaiee, V.; Alishahi, M. On the privacy protection of indoor location dataset using anonymization. Computers & Security. 2022, 17, 1-16.
- Gangarde, R.; Sharma, A.; Pawar, A.; Joshi, R.; Gonge, S. Privacy Preservation in Online Social Networks Using Multi-ple-Graph-Properties-Based Clustering to Ensure k-Anonymity, l-Diversity, and t-Closeness. Electronics. 2021, 10, 1-22.
Author action: According to the reviewer's comment, we have cited the literature on skew and similarity attacks through an extensive literature review, and we also have corrected the relevant errors. For details, please refer to the red words in Section 2 of the revised manuscript.
Reviewer#1, Concern #5:
- Skew and Similarity attacks can be better explained with a sample table data.
Author response: Thanks for the suggestion. We have added two sample table data to better explain the Skewness attacks and Similarity attacks in the revised manuscript.
Author action: According to the reviewer's comment, we have added two sample table data to better explain the Skewness attacks and Similarity attacks in the revised manuscript. For details, please refer to the red words in the introduction of the revised manuscript.
Reviewer#1, Concern #6:
- Related Works: This section should review the current state-of-the-art literature to formulate the research questions and to identify research gaps. But I feel this section just a summary or overview of different literature. There is no real analysis found. Perform more analysis. I suggest the authors to include a table that compares different techniques in the literature and their ways to handle different privacy attacks.
Author response: Thanks for the suggestion. We have added current state-of-the-art literature and added a table that compares different techniques in the literature and their ways of handling different privacy attacks. Moreover, we perform more analysis of all the literature.
Author action: According to the reviewer's comment, we have added a comparison table 4 of different privacy protection schemes and performed relevant analysis. For details, please refer to the red words in Section 2 of the revised manuscript.
Reviewer#1, Concern #7:
- In line no. 151, the authors summarised a research paper and wrote "equivalence class is less than the threshold value α" what is α here? Author must understand the context and use appropriate words not just the words from articles.
Author response: Thanks for the suggestion. Because our description is not very clear, the readers may not understand what the threshold value is. We have added the relevant contents and deleted the α in the revised manuscript according to the expert’s suggestions. The specific modification is as follows.
“Wang et al. [18] proposed two greedy clustering algorithms to partition equivalence classes to improve the privacy model and achieve data anonymity through generalization. The algorithms require that the frequency of sensitive data with high sensitivity in the same equivalent class is less than the threshold, but it cannot guarantee the security of sensitive data with low sensitivity.”
Author action: According to the reviewer's comment, we have revised the aforementioned errors in the revised manuscript. For details, please refer to the red words in Section 2 of the revised manuscript.
Reviewer#1, Concern #8:
- Multi-dimensional Data Preprocessing. In this section, there is no information on the data preprocessing. Report on the preprocessing used in this work.
Author response: Thanks for the suggestion. According to the expert’s suggestions, we have added the "Multi-dimensional Data Preprocessing" section in the "Schematic diagram of KAPP". In addition, we have updated the corresponding word description in "3.1. Multi-dimensional Data Preprocessing".
Author action: According to the reviewer's comment, we have updated the relevant content of “Multi-dimensional Data Preprocessing”. For details, please refer to the red words in Section 3.1 and figure 1 of the revised manuscript.
Reviewer#1, Concern #9:
- What is "sas" in line 240? Check all the equations for clarity and correctness. Some of the terms used in the equations are not explained.
Author response: Thanks for the suggestion. According to the expert’s suggestions, we have checked the notations of the whole paper and described the definition of sas.
sas represents the number of sensitive attributes.
Author action: According to the reviewer's comment, we have updated the relevant content and have double-checked the notations of the whole paper. For details, please refer to the red words in table 5 of the revised manuscript.
Reviewer#1, Concern #10:
- Algorithm 1 should use the equations presented above and show the step by step process to anonymise the data. But currently the algorithm is just short of sentences which are already written. I suggest the authors, write the algorithm with less words and more mathematically. Rewrite the algorithm considering my comments.
Author response: Thanks for the suggestion. According to the expert's suggestions, we have rewritten algorithm1 with fewer words and have made it more mathematical. We also have updated the description of algorithm 1.
Author action: According to the reviewer's comment, we have updated the relevant content in Algorithm 1. For details, please refer to the red words in Section 4 of the revised manuscript.
Reviewer#1, Concern #11:
- Security Analysis: This section should identify the potential challenges the proposed approach would face and how it can be tackled. It would be better if you can provide theorem and proof for various security analysis.
Author response: Thanks for the suggestion. According to the expert’s suggestions, we have updated the relevant content of security analysis with “theorem” and “proof”.
Author action: According to the reviewer's comment, we have updated the relevant content of the security analysis. For details, please refer to the red words in Section 5 of the revised manuscript.
Reviewer#1, Concern #12:
- The citations for diversity VD and Loss metrics are missing.
Author response: Thanks for the suggestion. According to the expert’s suggestions, we have added citations for diversity VD and Loss metrics. The specific modification is as follows.
The diversity VD reflects the distribution of sensitive data in the same equivalence class [29].
- Zhong, J.; Han, J.; Wang, H.; Chen, X. (k, l, e)-Anonymity: A resisting approximate attack model for sensitive attributes. Journal of Chinese Computer Systems. 2014, 5, 1491-1495.
The information loss rate reflects the availability of shared data after anonymity [10].
- Onesimu, J.; Karthikeyan, J.; Eunice, J.; Pomplun, M.; Dang, H. Privacy preserving attribute-focused anonymization scheme for healthcare data publishing. IEEE Access. 2022, 10, 86979-86997.
Author action: According to the reviewer's comment, we have added citations for diversity VD and Loss metrics. For details, please refer to the red words in Section 6.1 of the revised manuscript.
Reviewer#1, Concern #13:
- In figure 3 & 4 how anonymity percentage is calculated?
Author response: Thanks for the suggestion. Because our description is not very clear, readers may not know how to calculate the anonymity percentage. In the original manuscript, we gave the definition of anonymity and the word description for calculating the percentage of anonymity. However, we do not give a specific mathematical formula. In order to make the anonymous percentage clear to readers, we have added specific mathematical formula and explained the formula according to the reviewer's suggestion. The specific modification is as follows.
The anonymity reflects the privacy protection of shared data after anonymity [33]. It is defined as that attackers cannot successfully analyze the number of real sensitive data of specific individuals divided by the total number of data under skew and similarity attacks. The calculation formula is as follows.
(14)
where represents the total number of multi-dimensional data that all sensitive data of a sensitive attribute in the same equivalence class belong to the same parent node. represents the total number of multi-dimensional data whose proportion of a sensitive data in the same equivalence class is greater than the threshold %.
Author action: According to the reviewer's comment, we have added a detailed formula to calculate the anonymity percentage. For details, please refer to the red words in Section 6.1 of the revised manuscript.
Reviewer#1, Concern #14:
- Most of the privacy preserving study reports the anonymization process with respect to the change in the number of sensitive attributes. It is missing in this manuscript.
Author response: Thanks for the suggestion. According to the expert’s suggestions, we have updated the experimental simulation in section VI. We have added a new experiment to report the anonymization process of the number of sensitive attributes, and the results are shown in Figures 9-11.
Author action: According to the reviewer's comment, we have added new experiments whose results are shown in the figure, as well as theoretical analysis in the revised manuscript. For details, please refer to the red words in Section 6.2.3 of the revised manuscript.
Reviewer#1, Concern #15:
- Some of the de facto standard information loss and data utility metrics are not used in this work.
Author response: Thanks for the suggestion. We have updated the manuscript by revising the relevant content of the information loss and data utility metrics in the revised manuscript. In our work, since a generalization method is used to anonymize the original data in the proposed algorithms, we use a generalized loss metric [21] to compute the information loss.
- Wang, R.; Zhu, Y.; Chen T.; Chang, C. Privacy-preserving algorithms for multiple sensitive attributes satisfying t-closeness. Journal of Computer Science and Technology. 2018, 33, 1231-1242.
Author action: According to the reviewer's comment, we have updated the manuscript by revising the relevant content of the information loss and data utility metrics in the revised manuscript. For details, please refer to the red words in Section 6.1 of the revised manuscript.
Reviewer#1, Concern #16:
- Conclusion is not supported by results. No future work is available.
Author response: Thanks for the suggestion. We have updated the manuscript by revising the relevant content of conclusion in the revised manuscript.
Author action: According to the reviewer's comment, we have updated the manuscript by revising the relevant content of the conclusions in the revised manuscript. For details, please refer to the red words in Section 7 of the revised manuscript.
Reviewer#1, Concern #17:
- Some the recent references on privacy preserving studies are missing. The following may be considered.
Onesimu, J. Andrew, J. Karthikeyan, Jennifer Eunice, Marc Pomplun, and Hien Dang. "Privacy Preserving Attribute-Focused Anonymization Scheme for Healthcare Data Publishing." IEEE Access 10 (2022): 86979-86997.
Jayapradha, J., M. Prakash, Youseef Alotaibi, Osamah Ibrahim Khalaf, and Saleh Ahmed Alghamdi. "Heap Bucketization Anonymity—An Efficient Privacy-Preserving Data Publishing Model for Multiple Sensitive Attributes." IEEE Access 10 (2022): 28773-28791.
Ni, Chunchun, Li Shan Cang, Prosanta Gope, and Geyong Min. "Data anonymization evaluation for big data and IoT environment." Information Sciences 605 (2022): 381-392.
Author response: Thanks for the suggestion. We have updated the manuscript by revising the relevant content of references. Specially, we have cited the three references mentioned above and other recent references.
Author action: According to the reviewer's comment, we have cited the three references mentioned above and other recent references. For details, please refer to the red words in the references of the revised manuscript.
Reviewer 2 Report
In this paper, the authors studied the multi-dimensional data K-anonymity privacy protection problems and proposed a K- anonymity privacy protection algorithm called KAPP. Specifically, the authors first summarized the clustering methods of anonymous data. Then, based on the improved African vulture optimization and quasi-identifier data distance, the authors realized the clustering of the multi-dimensional data and initialization of the equivalence classes. Finally, the experimental results showed the effectiveness of the proposed KAPP algorithm.
This is a very detailed article. However, there are some issues that the authors may consider.
1. In the abstract, the authors directly proposed the basic details of the KAPP scheme. The authors should explain the motivation of this work by briefly introducing the background of the research topic.
2. In the introduction, the authors introduced the background of the research topic and proposed several problems that need to be considered in this paper. The way of narration is very detailed. However, the summary of the contributions seems redundant. Thus, it may be nonintuitive for readers to find the main contributions.
3. In the section “Algorithm Principle”, the authors proposed the detailed workflow of the KAPP algorithm and presented a figure to provide a vivid narration. However, a figure full of words may not let people understand the KAPP algorithm process intuitively.
4. There are too many notations in Section â…£. A notation table may be a good choice.
5. There are many formulas and figures in the section “Algorithm Principle” and “Experimental Simulation”. This makes the layout of the article not in good order. Specifically, the line distance is inconsistent, and large blank areas appear on the page.
6. There are few cryptographic proofs in the “Security Analysis” section, making it hard to convince the readers of the security guarantee of the algorithm. Writing “Definitions” and “Proofs” may be a good choice.
Author Response
Comments of Reviewer#2:
Summary
In this paper, the authors studied the multi-dimensional data K-anonymity privacy protection problems and proposed a K- anonymity privacy protection algorithm called KAPP. Specifically, the authors first summarized the clustering methods of anonymous data. Then, based on the improved African vulture optimization and quasi-identifier data distance, the authors realized the clustering of the multi-dimensional data and initialization of the equivalence classes. Finally, the experimental results showed the effectiveness of the proposed KAPP algorithm. This is a very detailed article.
Author response: Thank the expert for affirming our paper.
Author action: Thank the expert for affirming our paper.
Reviewer#2, Concern #1:
- In the abstract, the authors directly proposed the basic details of the KAPP scheme. The authors should explain the motivation of this work by briefly introducing the background of the research topic.
Author response: Thanks for the suggestion. According to the reviewer's comment, we have rewritten the abstract of the paper. The revised abstract explains the motivation of this work by briefly introducing the background of the research topic.
Author action: According to the reviewer's comment, we have updated the manuscript by revising the content of the abstract in the revised manuscript. For details, please refer to the abstract of the revised manuscript.
Reviewer#2, Concern #2:
- In the introduction, the authors introduced the background of the research topic and proposed several problems that need to be considered in this paper. The way of narration is very detailed. However, the summary of the contributions seems redundant. Thus, it may be nonintuitive for readers to find the main contributions.
Author response: Thanks for the suggestion. According to the reviewer's comment, we have updated the summary of the contributions.
Author action: According to the reviewer's comment, we have updated the manuscript by revising the summary of the contributions in the revised manuscript. For details, please refer to the red words of the summary of the contributions in the introduction of the revised manuscript.
Reviewer#2, Concern #3:
- In Section “Algorithm Principle”, the authors proposed the detailed workflow of the KAPP algorithm and presented a figure to provide a vivid narration. However, a figure full of words may not let people understand the KAPP algorithm process intuitively.
Author response: Thanks for the suggestion. According to the reviewer's comment, we have updated the KAPP schematic diagram with more concise words.
Author action: According to the reviewer's comment, we have updated the manuscript by revising Figure 1 in the revised manuscript. For details, please refer to Figure 1 of the revised manuscript.
Reviewer#2, Concern #4:
- There are too many notations in Section â…£. A notation table may be a good choice.
Author response: Thanks for the suggestion. According to the reviewer's comment, we have added a notation table, which helps simplify the notations in Section â…£.
Author action: According to the reviewer's comment, we have added a notation table to simplify the notations in Section â…£. For details, please refer to the red words in Section 3 of the revised manuscript.
Reviewer#2, Concern #5:
- There are many formulas and figures in Section “Algorithm Principle” and “Experimental Simulation”. This makes the layout of the article not in good order. Specifically, the line distance is inconsistent, and large blank areas appear on the page.
Author response: Thanks for the suggestion. According to the expert's suggestions, we have adjusted the position of the formulas in the "algorithm principle" and the position of the figures in the "experimental simulation" to reduce the blank areas as much as possible.
Author action: According to the reviewer's comment, we have adjusted the position of the formulas and the figures. For details, please refer to the formulas and the figures in Section 3 and Section 6 of the revised manuscript.
Reviewer#2, Concern #6:
- There are few cryptographic proofs in the “Security Analysis” section, making it hard to convince the readers of the security guarantee of the algorithm. Writing “Definitions” and “Proofs” may be a good choice.
Author response: Thanks for the suggestion. According to the expert’s suggestions, we have updated the relevant content of security analysis with “theorem” and “proof”.
Author action: According to the reviewer's comment, we have revised the relevant content of the security analysis. For details, please refer to the red words in Section 5 of the revised manuscript.
Reviewer 3 Report
The authors propose a multi-dimensional sensitive data clustering algorithm based on improved African vulture optimization. In particular, they improve the initialization of the cluster center, the fitness calculation method, and the solution update strategy. The authors also exploit an equivalence class partition, based on the sensitive data distribution difference, to achieve the safe partition of equivalence classes. Additionally, their proposal generalizes quasi-identifier data and sensitive data with significant weights. Finally, the authors present experimental results by showing that their proposal improves the clustering accuracy, diversity, and anonymity under skew and similarity attacks. The research topic addressed by the authors is fascinating, but their work needs improvements; some suggestions are reported in the following.
Abstract
The abstract needs to be more detailed. In particular, it is difficult to understand the addressed problem and its proposed solution. Therefore, it is kindly suggested to improve the abstract by better describing the problem and the presented solution.
Introduction
The Introduction section needs be improved. In particular, the storyline that the authors want to define is not linear. It is kindly suggested that the authors better define the problem and better describe the aim of the proposed research. Additionally, when the authors introduce equivalent classes, they could consider introducing other methodologies that compute equivalent classes exploiting similarity functions to guarantee k-anonymity as defined in the following articles:
- Caruccio L, et al. A decision-support framework for data anonymization with application to machine learning processes, Information Sciences, Elsevier, Vol. 613 (2022).
- Yan Y, et al. A weighted k-member clustering algorithm for k-anonymization, Computing, Springer, Vol. 103, (2021).
- Lin J.L, et al. An efficient clustering method for k-anonymization, in Proceedings of the 2008 international workshop on Privacy and anonymity in information society, March 2008, Pages 46–50, https://doi.org/10.1145/1379287.1379297.
Moreover, it is kindly suggested that the authors shorten the list of main contributions and add a brief background of the other sections at the end of the introduction.
Related Work
The Related Work section is a simple list of shortly described articles. It is kindly suggested that the authors categorize them by improving their descriptions and adding pros and cons for each cited article. Additionally, it is kindly suggested that the authors consider adding a table to classify the works w.r.t. criteria of interest, such as the approach used, privacy guarantee, anonymization technique, data utility metrics, and so on.
Section 3 contains a fragmented sentence -> "The details are as ". Please check e fix it.
The end of section 3.2.2 contains a typo -> "where it represents" should have a capital letter.
In Section 3.2.3, the authors write: "is okay" please consider replacing it to be more formal.
It is kindly suggested that the authors check all formulas in section 3 by improving their descriptions because they are hard to understand.
Section 4 needs be better detailed, mainly if the authors aim to describe the algorithm's pseudo-code.
In Section 6, the dataset used for experimental evaluation is briefly described. It is kindly suggested that the authors improve the dataset description and add the data sources. Additionally, all formulas need to be checked, and their description improved.
A discussion section could be added before the conclusion section. In particular, the authors should discuss their contributions through salient points.
The conclusion section is brief, and no future directions are reported. It is kindly suggested that the authors improve the conclusion section description and add future directions for their work.
Finally, typos and spelling English checks are required.
Author Response
Reviewer#3:
Summary
The authors propose a multi-dimensional sensitive data clustering algorithm based on improved African vulture optimization. In particular, they improve the initialization of the cluster center, the fitness calculation method, and the solution update strategy. The authors also exploit an equivalence class partition, based on the sensitive data distribution difference, to achieve the safe partition of equivalence classes. Additionally, their proposal generalizes quasi-identifier data and sensitive data with significant weights. Finally, the authors present experimental results by showing that their proposal improves the clustering accuracy, diversity, and anonymity under skew and similarity attacks. The research topic addressed by the authors is fascinating, but their work needs improvements; some suggestions are reported in the following.
Author response: Thank the expert for affirming our paper.
Author action: Thank the expert for affirming our paper.
Reviewer#3, Concern #1:
- The abstract needs to be more detailed. In particular, it is difficult to understand the addressed problem and its proposed solution. Therefore, it is kindly suggested to improve the abstract by better describing the problem and the presented solution.
Author response: Thanks for the suggestion. According to the reviewer's comment, we have rewritten the abstract of the paper. The revised abstract can explain the motivation of this work by briefly introducing the background of the research topic.
Author action: According to the reviewer's comment, we have updated the manuscript by revising the content of the abstract in the revised manuscript. For details, please refer to the abstract of the revised manuscript.
Reviewer#3, Concern #2:
- The Introduction section needs be improved. In particular, the storyline that the authors want to define is not linear. It is kindly suggested that the authors better define the problem and better describe the aim of the proposed research. Additionally, when the authors introduce equivalent classes, they could consider introducing other methodologies that compute equivalent classes exploiting similarity functions to guarantee k-anonymity as defined in the following articles:
- Caruccio L, et al. A decision-support framework for data anonymization with application to machine learning processes, Information Sciences, Elsevier, Vol. 613 (2022).
- Yan Y, et al. A weighted k-member clustering algorithm for k-anonymization, Computing, Springer, Vol. 103, (2021).
- Lin J.L, et al. An efficient clustering method for k-anonymization, in Proceedings of the 2008 international workshop on Privacy and anonymity in information society, March 2008, Pages 46–50, https://doi.org/10.1145/1379287.1379297.
Author response: Thanks for the suggestion. According to the expert's suggestions, we have updated the Introduction section, including the description of the problem and the aim of the proposed research. Moreover, we have cited the literature in related work.
Author action: According to the reviewer's comment, we have updated the description of the problem and the aim of the proposed research. Moreover, we have cited the literature. For details, please refer to the red words in Section 1 of the revised manuscript.
Reviewer#3, Concern #3:
- Moreover, it is kindly suggested that the authors shorten the list of main contributions and add a brief background of the other sections at the end of the introduction.
Author response: Thanks for the suggestion. According to the expert's suggestions, we have shorted the list of main contributions and have added a brief background of the other sections at the end of the introduction.
Author action: According to the reviewer's comment, we have shorted the list of main contributions and added a brief background of the other sections. For details, please refer to the red words in the introduction of the revised manuscript.
Reviewer#3, Concern #4:
- The Related Work section is a simple list of shortly described articles. It is kindly suggested that the authors categorize them by improving their descriptions and adding pros and cons for each cited article. Additionally, it is kindly suggested that the authors consider adding a table to classify the works w.r.t. criteria of interest, such as the approach used, privacy guarantee, anonymization technique, data utility metrics, and so on.
Author response: Thanks for the suggestion. According to the expert's suggestions, we have revised the description of all the literature and have classified them. Additionally, we have added a table that compares different techniques in the literature and their ways of handling different privacy attacks.
Author action: According to the reviewer's comment, we have revised the description of all the literature as well as classified them, and we have added a table that compares different techniques in the literature and their ways of handling different privacy attacks. For details, please refer to the red words in Section 2 of the revised manuscript.
Reviewer#3, Concern #5:
- Section 3 contains a fragmented sentence -> "The details are as ". Please check e fix it.
Author response: Thanks for the suggestion. According to the expert's suggestions, we have revised the aforementioned errors in the revised manuscript.
Author action: According to the reviewer's comment, we have corrected the relevant errors. For details, please refer to the red words in Section 3 of the revised manuscript.
Reviewer#3, Concern #6:
- The end of section 3.2.2 contains a typo -> "where it represents" should have a capital letter.
Author response: Thanks for the suggestion. According to the expert's suggestions, we have revised the aforementioned typos in the revised manuscript.
Author action: According to the reviewer's comment, we have corrected the relevant errors. For details, please refer to the red words in Section 3.2.2 of the revised manuscript.
Reviewer#3, Concern #7:
- In Section 3.2.3, the authors write: "is okay" please consider replacing it to be more formal.
Author response: Thanks for the suggestion. According to the expert's suggestions, we have revised it in the revised manuscript.
Author action: According to the reviewer's comment, we have replaced it with more formal sentences. For details, please refer to the red words in Section 3.2.3 of the revised manuscript.
Reviewer#3, Concern #8:
- It is kindly suggested that the authors check all formulas in section 3 by improving their descriptions because they are hard to understand.
Author response: Thanks for the suggestion. According to the expert's suggestions, we have double-checked all formulas in section 3 and have revised their description of the formulas to make them easier to understand.
Author action: According to the reviewer's comment, we have double-checked all formulas in section 3 and have revised their description of the formulas to make them easier to understand. For details, please refer to the red words in Section 3 of the revised manuscript.
Reviewer#3, Concern #9:
- Section 4 needs be better detailed, mainly if the authors aim to describe the algorithm's pseudo-code.
Author response: Thanks for the suggestion. According to the expert's suggestions, we have rewritten algorithm 1 in Section 4 with fewer words and made it more mathematical. We also have updated the description of algorithm 1.
Author action: According to the reviewer's comment, we have updated the relevant content in Algorithm 1. For details, please refer to the red words in Section 4 of the revised manuscript.
Reviewer#3, Concern #10:
- In Section 6, the dataset used for experimental evaluation is briefly described. It is kindly suggested that the authors improve the dataset description and add the data sources. Additionally, all formulas need to be checked, and their description improved.
Author response: Thanks for the suggestion. According to the expert's suggestions, we have revised the dataset description and have added the data sources in section 6.1 of the revised manuscript. Additionally, we have checked all formulas and have revised their description in section 6.1 of the revised manuscript.
Author action: According to the reviewer's comment, we have revised the dataset description and have added the data sources in the revised manuscript. Additionally, we have checked all formulas and have revised their description in Section 6.1 of the revised manuscript. For details, please refer to the red words in Section 6.1 of the revised manuscript.
Reviewer#3, Concern #11:
- A discussion section could be added before the conclusion section. In particular, the authors should discuss their contributions through salient points.
Author response: Thanks for the suggestion. According to the expert's suggestions, we have revised the conclusion section in the revised manuscript.
Author action: According to the reviewer's comment, we have revised the conclusion section in the revised manuscript. For details, please refer to the red words in Section 7 of the revised manuscript.
Reviewer#3, Concern #12:
- The conclusion section is brief, and no future directions are reported. It is kindly suggested that the authors improve the conclusion section description and add future directions for their work.
Author response: Thanks for the suggestion. We have updated the manuscript by revising the relevant content of the conclusion in the revised manuscript.
Author action: According to the reviewer's comment, we have updated the manuscript by revising the relevant content of conclusions in the revised manuscript. For details, please refer to the red words in Section 7 of the revised manuscript.
Reviewer#3, Concern #13:
- Finally, typos and spelling English checks are required.
Author response: Thanks for the suggestion. According to the expert's suggestions, we have double-checked the spelling English of the whole paper and have revised the typos.
Author action: According to the reviewer's comment, we have double-checked the spelling English of the whole paper and have revised the typos.
Round 2
Reviewer 1 Report
I appreciate the authors effort in addressing my concerns. However, the Authors have partially addressed my comments. The following are the comments that are not addressed and I could find the responses. I found the author starting their responses only from comment 5. If this is by mistake please check and submit the response and authors' action as a separate files. Otherwise I feel the manuscript has improved a lot from the previous version.
Previous comments that are not addressed.
Abstract:
First line of the abstract doesn't form a correct sentence. Rephrase it.
Abstract should clearly define the problem statement. The authors have mentioned multiple techniques to improve the clustering techniques, equivalence class partition and so on. But the main objective for doing this is not clear.
What is the role of African vulture optimization in this research can be briefed.
How the experimental results are evaluated? Performance details metrics required.
How the proposed approach is thwarting the skew and semantic attacks?
- Considering the above queries, please rewrite the abstract for better clarity.
Introduction:
The first two paragraphs are mainly focused on multi-dimensional data protection however, multi-dimensional data is not introduced. Introduce it with a sample table or records.
In sensitive attribute protection, there are single sensitive attribute protection and multiple sensitive attribute protection. Authors must clearly specify the type of problem they are addressing.
Line 62 is not true. There is a huge amount of literature study available. The authors must identify and cite literature that works on skew and similarity attacks.
Why not l-diversity, t-closeness, and their relevant study not mentioned in this section? These techniques are mainly proposed to address Skewness attacks and similarity attacks. There are much more like bucketization, and so on available.
Authors must explore the current state-of-the-art techniques and identify the research gap to propose the solution. A mere statement like "less study" is not acceptable when there is more literature already available.
Reviewer 2 Report
I am fine with accepting this manuscript.
Author Response
We are very grateful to the third reviewer for his careful review of the paper and his valuable suggestions.
Reviewer 3 Report
The authors did not address the previous remarks. In particular, their proposal remains difficult to understand, and their anonymity model (i.e., k-anonymity) is susceptible to reidentification attacks. In this light, the authors' proposal also remains weak because other anonymity models, such as l-diversity, t-closeness and differential privacy, can preserve privacy by overcoming the limitations of the k-anonymity model. Moreover, the authors never motivate the choice of using k-anonymity w.r.t. the other models. For example, Why use k-anonymity as a privacy model instead of differential privacy?
In what follows, some additional concerns.
Abstract
The abstract remains difficult to understand. In particular, it needs to be clarified the application scenario that the authors want to analyze. Also, the idea underlying the solution provided by the authors needs to be revised.
Introduction
The storyline the authors have defined in the introduction is totally unclear. In particular, the problem definition and the aim of the authors' proposal are never defined. The tables in the introduction section are redundant. The text from line 92 to line 108 is entirely unclear and difficult to understand. The list of main contributions needs to be shortened; it contains too many details that are not useful for comprehension. Additionally, the authors did not broaden the bibliography as suggested.
Related Work
The Related Work section remains a simple list of shortly described articles. The authors did not categorize them, and their descriptions are not detailed. Additionally, table 4 contains few papers and differential privacy is never mentioned as a privacy model despite being the model that offers more guarantees. A precise comparison table was defined in one of the papers suggested in my previous review (The authors could refer to that reported in article 1). Moreover, the bibliography is not examined in detail. Differential privacy methodologies are never exploited in any cited articles, even if such methodology is robust to reidentification attacks.
The formulas in section 3 remain challenging to understand.
Section 4 remains not detailed.
In Section 6, the dataset used for experimental evaluation remains superficially described together with all formulas.
A discussion section has not been added before the conclusion section, and the authors' contributions through salient points have not been added.
The conclusion section remains brief, and also the future directions are superficially described.
The quality of English is also degraded.
Round 3
Reviewer 3 Report
Although authors made a further effort to improve their paper, having revised it in one week, starting from a reject response they couldn’t reasonably fix all the problems that I raised in my last review.
In particular, in order to put the paper in an acceptable quality, the following problems still remain to be tackled:
Abstract
It is kindly suggested that the authors better motivate the fact that the privacy reference model is t-closeness and not k-anonymity. The k-anonymity concept is exploited to obtain the satisfiability of the
t-closeness model. This aspect of the abstract is difficult to understand.
Introduction
It is kindly suggested that the authors better motivate the choice of using the t-closeness model by emphasizing how its privacy properties may safeguard against skewness and similarity attacks.
Conclusion
As a future direction for the author's proposal, consider integrating the methodology in data analytics and machine learning contexts.
Finally, further proofreading and grammar English checks are still required.
